# Regulatory T cells expressing CD19-targeted chimeric antigen receptor restore homeostasis in Systemic Lupus Erythematosus

M. Doglio [1] ✉, A. Ugolini[1], C. Bercher-Brayer[1], B. Camisa[1], C. Toma[1], R. Norata[2], S. Del Rosso[3], R. Greco [4], F. Ciceri [4], F. Sanvito [2,5], M. Casucci [6], A. A. Manfredi [7] & C. Bonini [1] ✉

Systemic Lupus Erythematosus (SLE) is a progressive disease leading to immune-mediated tissue damage, associated with an alteration of lymphoid organs. Therapeutic strategies involving regulatory T (Treg) lymphocytes, which physiologically quench autoimmunity and support long-term immune tolerance, are considered, as conventional treatment often fails. We describe here a therapeutic strategy based on Tregs overexpressing FoxP3 and harboring anti-CD19 CAR (Fox19CAR-Tregs). Fox19CAR-Tregs efficiently suppress proliferation and activity of B cells in vitro, which are relevant for SLE pathogenesis. In an humanized mouse model of SLE, a single infusion of Fox19CAR-Tregs restricts autoantibody generation, delay lymphopenia (a key feature of SLE) and restore the human immune system composition in lymphoid organs, without detectable toxicity. Although a short survival, SLE target organs appear to be protected. In summary, Fox19CAR-Tregs can break the vicious cycle leading to autoimmunity and persistent tissue damage, representing an efficacious and safe strategy allowing restoration of homeostasis in SLE.

Systemic Lupus Erythematous (SLE) is characterized by the aberrant immune response towards nuclear antigens with an inflammatory reaction against healthy tissues. Auto-antibodies against nuclear antigens are the hallmarks of this condition and form immune complexes by binding to nucleic acids that deposit in tissues fostering the further activation of the immune system[1,2]. Follicular hyperplasia is a common finding in lymph nodes and spleen of SLE patients[3,4]. All renal compartments are involved by the inflammatory process in lupus nephritis (LN), with a great variability between patients and different histological

manifestations[5,6]. SLE severity correlates with changes in lymphoid organs, which reflect chronicity of the autoimmune response, and inflammatory and degenerative changes in the target organs such as the kidney, with the deposition of fibrous tissue and the progressive disruption of the organ architecture[7].

Standard medical treatment is based on steroids and immunosuppressive agents, with side effects and frequent disease flares at their discontinuation[8]. There had been no significant improvement in the outcome of LN in the three decades to 2011[9], indicating that

[1]Experimental Hematology Unit, Division of Immunology Transplantation and Infectious Diseases (DITID), IRCCS San Raffaele Scientific Institute, Milan, Italy. [2]GLP Test Facility, San Raffaele Telethon Institute for Gene Therapy (SR-Tiget), IRCCS San Raffaele Scientific Institute, Milan, Italy. [3]Autoimmunity Lab, IRCCS San Raffaele Hospital, Milan, Italy. [4]Hematology and Bone Marrow Transplantation Unit, IRCCS San Raffaele Hospital, Milan, Italy. [5]Pathology Unit, Division of Experimental Oncology, IRCCS San Raffaele Scientific Institute, Milan, Italy. [6]Innovative Immunotherapies Unit, Division of Immunology Transplantation and Infectious Diseases (DITID), IRCCS San Raffaele Scientific Institute, Milan, Italy. [7]Autoimmunity and Vascular Inflammation Unit, Division of Immunology Transplantation and Infectious Diseases (DITID), IRCCS San Raffaele Scientific Institute, Milan, Italy. ✉e-mail: doglio.matteo@hsr.it; bonini.chiara@hsr.it

approaches based on conventional immunosuppressive agents are unlikely to yield additional relevant beneficial effects in these patients. New treatments, including biologic therapies, have so far produced suboptimal results with several agents that have not met their primary endpoints in clinical trials[10–12]. Therefore, SLE represents an important unmet clinical need and new strategies should be found to add to the few available therapies with proven efficacy[13].

Regulatory T cells (Tregs) are CD4+ T lymphocytes endowed with immune suppressive capacities that maintain immune homeostasis and prevent excessive inflammatory responses[14]. Tregs have been largely studied in the context of autoimmune diseases and a reduction both in terms of numbers and function contributes to the SLE pathogenesis[15]. Since their discovery, Tregs have represented attractive candidates for the treatment of autoimmune diseases due to their immunomodulatory properties[16]. In the context of hematopoietic stem cell transplantation (HSCT) for autoimmune diseases, including SLE[17], the restoration of the Treg compartment correlates with long-term disease remission. Clinical trials have been conducted with expanded polyclonal Tregs, which proved to be safe but resulted in unsatisfactory clinical improvements[18], possibly due to the low amount of antigen-specific regulatory cells in the infused product[19].

Chimeric Antigen Receptors (CARs) are chimeric molecules capable of redirecting the specificity of engineered cells against target antigens, while simultaneously boosting their activation[20,21]. Conventional T cells (Tconvs) expressing CARs had striking clinical results in patients affected by hematological malignancies, since their cytotoxic action could be specifically redirected against transformed cells[22]. CD19-CAR Tconvs proved effective in pre-clinical mouse models and in patients with refractory SLE[23–25] due to a deep B-cell depletion followed by B-cell repopulation over time. However, CARs also represent an ideal solution for the generation of disease-relevant antigen-specific Tregs, thanks to the possibility of enhancing their immunosuppressive functions independently of conventional actions such as cytotoxicity. Efficacy of CAR-Tregs have been shown in preclinical models of type I diabetes and solid organ rejection[26,27]. Imura et al. recently demonstrated the efficacy of naïve-derived anti-CD19 CAR-Tregs in controlling B cell activation and improving Graft-versus-Host Disease (GvHD) in a xenograft mouse model[28].

Here we develop and investigate a CAR-Treg cellular product for the treatment of SLE. Engineered cells overexpressing FoxP3 and harboring an anti-CD19 second-generation CAR prove effective in controlling B cell activation in vitro and in restoring the human immune system composition in lymphoid organs in an ad hoc humanized mouse model of SLE. This approach represents a promising therapeutic strategy to control refractory SLE patients, who often experience poor disease control with conventional treatments.

## Results

### Engineered 19CAR-Tregs maintain immunomodulatory properties and acquire antigen specificity

Considering the relevant role of B cells, autoantibodies and immune complexes in lupus pathogenesis, we resolved to generate anti-CD19 CAR-Tregs to control self-reactive B lymphocytes. We developed a bi-directional lentiviral (LV) vector encoding for a second-generation anti-CD19 CAR constituted by a spacer region derived from the extracellular domain of the human low-affinity nerve growth factor receptor (LNGFR)[29], the CD3z chain, and the intracellular portion of CD28 (CAR19.28z LV, Suppl. Fig. 1a), the most active co-stimulatory domain for the CAR-Treg function[30].

To generate anti-CD19 CAR-Tregs (19CAR-Tregs), we sorted CD4+CD25+ cells from healthy donors' (HD) peripheral blood mononuclear cells (PBMCs) and we expanded them in the presence of IL-2 and rapamycin[31]. We optimized the protocol for Treg expansion to include lentiviral transduction for CAR-Treg generation (Suppl. Fig. 1b). Mean transduction efficiency, assessed by flow cytometry, was

39.4% ± 12.0%, with stable co-expression of the transgenes for up to 21 days of culture (Suppl. Fig. 1c–e). 19CAR-Tregs and untransduced (UT) cells displayed a similar expansion rate (43.7 fold ± 11.1 for 19CAR-Tregs and 39.4 ± 14.7 for UT-Tregs) at days 14 and purity (median CD4+CD25+CD127−: 84.4% and 70,2% for 19CAR-Tregs and 89,7% and 67,6% for UT-Tregs) at days 14 and 21 after the initial stimulation (Suppl. Fig. 1f, g), indicating that the lentiviral transduction impacts neither on Treg expansion nor on cell phenotype.

To verify the 19CAR-Treg suppressive capacities, we co-cultured autologous PBMCs with either 19CAR-Tregs or UT-Tregs in the presence of anti-CD3/anti-CD28 stimulation beads. After 7 days, we measured the proliferation of PBMCs by flow cytometry and observed a comparable suppressive capacity in 19CAR-Tregs and UT-Tregs, with a suppression index of 83.0% ± 7.7% for UT-Tregs and 74.0% ± 19.3% for 19CAR-Tregs (Suppl. Fig. 1h). These results indicate that the lentiviral transduction and the culture do not alter Treg function.

We then co-cultured autologous B lymphocytes with either 19CAR-Tregs or UT-Tregs. Briefly, we activated B cells, by challenging them with irradiated CD40L-transduced 3T3 cells. 19CAR-Tregs, but not UT-Tregs, suppressed the proliferation of autologous B lymphocytes (mean B cell proliferation: 32.9% B cells alone, 30.4% in UT- and 12,6% in 19CAR-Tregs with suppression indexes of 0% and 61,2%, respectively), highlighting the effectiveness with which CAR molecules redirect the suppressive activity in an antigen-specific manner (Suppl. Fig. 1i).

In chronic inflammatory conditions, Tregs can be reprogrammed towards an effector phenotype[32]. To exclude that the reduced B lymphocyte proliferation might be due to a CAR-Treg reprogramming and consequent B-cell killing, we co-cultured ALL-CM, a CD19+ cell line, with either 19CAR-Tregs, 19CAR-Tconvs or UT-Tconvs. After 3 days, 19CAR-Tconvs killed target cells, while 19CAR-Tregs and UT-Tconvs equally spared CD19+ targets (p-value < 0.01) (Suppl. Fig. 1j–l). Overall, the results indicate that CAR-Tregs can be generated and propagated with relative ease and that engineered Tregs retain their suppressive capacities, which unravel upon recognition of their specific antigens.

### Fox19CAR-engineered Tregs maintain immunomodulatory properties and show superior antigen-specific suppressive functions

The adoptive transfer of Tregs requires an extensive validation of their safety profile, since they might undergo reprogramming toward conventional Th17 cells[32], and Tconvs might contaminate the final cellular product[33]. Since FoxP3 over-expression redirects inflammatory phenotypes toward a suppressive function[34], we decided to stabilize the Tregs through the constitutive up-regulation of FoxP3, by developing a LV encoding for the anti-CD19 CAR and *FoxP3* genes separated by a thosea asigna virus 2 A peptide (T2A) and under the control of a PGK promoter (Fox19CAR). As control, we employed a LV encoding for the *FoxP3* gene alone, devoid of the CAR construct (FoxP3 LV) (Fig. 1a).

We transduced sorted CD4+CD25+ bona fide Treg cells with the developed constructs according to the protocol described above. Transduction efficiency, measured at day 14, was higher for Fox19CAR and FoxP3 LV than for CAR19.28z LV (mean: 72.5% ± 12.3%; 91.1% ± 1.5%; 46.1% ± 12,5% respectively; Fox19CAR and FoxP3 LV vs. CAR19.28z LV p-value < 0.001) (Fig. 1b). Fox19CAR LV transduction promoted robust co-localization of CAR and FoxP3, indicating that both transgenes were actively transcribed (Fig. 1c). CAR expression was similar in 19CAR- and Fox19CAR-Tregs (Suppl. Fig. 2a). We did not find significant differences in terms of expansion between Fox19CAR-, FoxP3-, 19CAR- and UT-Tregs (mean Fold Expansion at day 14: 41.8 ± 5.2; 28.1 ± 8.0; 43.3 ± 5.2 and 41.0 ± 12.0 respectively Fig. 1d). Finally, the Treg purity in the cellular products after 14 days of culture was identical in the four groups. (Suppl. Fig. 2b).

In a 7 day polyclonal suppression assay we observed that Fox19CAR-Tregs suppressed the proliferation of activated PBMCs as

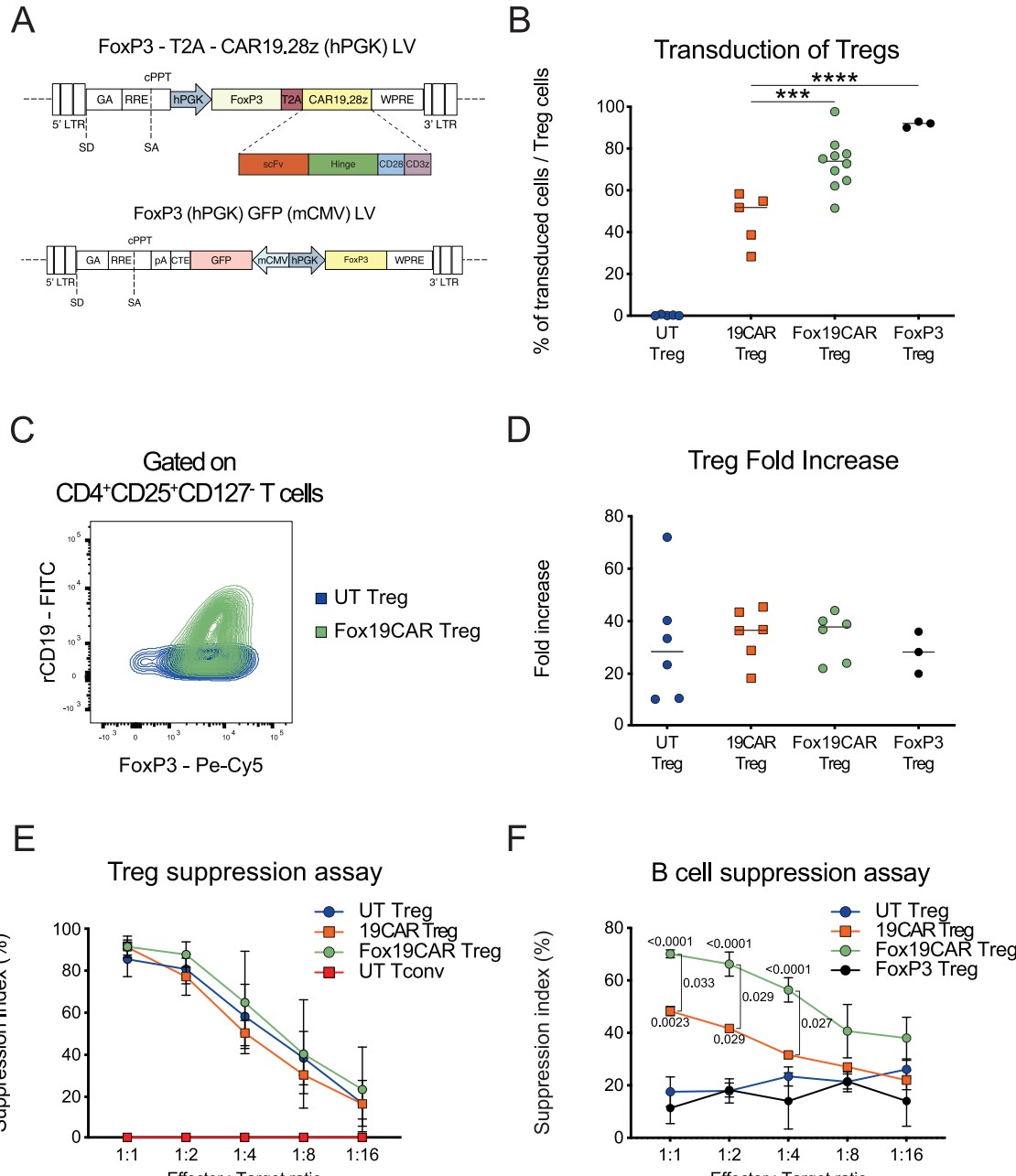

**Fig. 1 | Fox19CAR-Treg in vitro generation and validation. A** Fox19CAR lentiviral vector (LV) and FoxP3 LV schematic representations. The *FoxP3* gene and an anti-CD19 second-generation CAR are inserted in an unidirectional LV and under the control of a Phosphoglycerate Kinase (PGK) promoter. The vector encoding for the *FoxP3* gene alone is a bi-directional LV with the *FoxP3* gene in sense (PGK promoter) and the enhanced Green Fluorescent Protein (eGFP) in antisense (minimal CMV or mCMV promoter). LTR long terminal repeat, SD splice donor, SA splice acceptor, GA gag-pol element, RRE REV responsive element, cPPT central polypurine tract, pA polyadenilation signal, CTE constitutive transport element, WPRE woodchuck hepatitis virus post-transcriptional regulatory element. **B** Transduction efficiency of CAR19.28z LV, Fox19CAR LV and FoxP3 LV measured as percentage of either CAR+ or GFP+ cells among CD4+CD25+CD127-FoxP3+ lymphocytes at day +14. Recombinant CD19 for UT-, 19CAR- and Fox19CAR-Tregs and GFP for FoxP3 LV respectively, were employed to assess the transduction percentage. $N = 5$ for UT- and 19CAR-Tregs. $N = 10$ for Fox19CAR-Tregs. $N = 3$ for FoxP3-Tregs. One-way ANOVA with Tukey correction for multiple comparison. ***$p$-value 0.0008,

****$p$-value < 0.0001. **C** Representative flow cytometry plot for co-localization of the CAR construct and FoxP3 in Fox19CAR-Tregs compared to untransduced (UT) cells. CAR expression was assessed using the human rCD19. **D** Expansion rate of UT-, 19CAR-, Fox19CAR- and FoxP3-Tregs, assessed at day +14 since the initial stimulation. $N = 6$ for UT-, Fox19CAR- and 19CAR-Tregs. $N = 3$ for FoxP3-Tregs. One-way ANOVA with Tukey correction for multiple comparison. **E** Polyclonal suppressive capacities of UT-, 19CAR- and Fox19CAR-Tregs. Results are expressed as Suppression Index, calculated as Suppression index = [1-(PBMCs' proliferation with Tregs)/(PBMCs' proliferation alone)] * 100. $N = 3$ for UT-, Fox19CAR- and 19CAR-Tregs. $N = 3$ for UT conventional T cells. Two-way ANOVA with Tukey correction for multiple comparison. **F** Antigen-specific suppressive capacities of UT-, 19CAR-, Fox19CAR- and FoxP3-Tregs. Results are expressed as Suppression index = [1-(B cell proliferation with Tregs)/(B cell proliferation alone)] * 100. $N = 6$ for UT- and Fox19CAR-Tregs. $N = 3$ for 19CAR- and FoxP3-Tregs. Two-way ANOVA with Tukey correction for multiple comparison. The exact $p$-values are reported in the graph. All the results are expressed as mean ± standard deviation.

efficiently as 19CAR- and UT Tregs (Fig. 1e) and when cultured together with autologous B lymphocytes, Fox19CAR engineered cells not only showed a potent antigen-specific suppression but also outperformed 19CAR-Tregs in controlling the B cell proliferation (Fox19CAR- vs FoxP3- and UT-Tregs p-value < 0.0001 at 1:1 and 1:4 ratios; Fox19CAR- vs 19CAR-Tregs p-value < 0.05 at 1:1 and 1:4 ratios) (Fig. 1f). Overall, Fox19CAR-Tregs retain their suppressive capacities and display superior antigen-specific suppression than 19CAR-Tregs.

Naïve-derived (CD4 + 25 + CD127-CD45RA+ cells) Tregs have been recently employed to generate CAR-Tregs[28]. Our protocol, based on the Fox19CAR LV and use of rapamycin in culture to transduce CD4 + CD25+ sorted Tregs, could represent a valuable alternative approach. We thus compared Fox19CAR-Tregs with naïve-derived anti-CD19 CAR-Tregs. We firstly isolated CD4 + CD25+ cells from HD PBMCs with magnetic cell separation. Sorted cells were divided in two fractions: one was transduced with the Fox19CAR LV and cultured with rapamycin, whereas the second fraction was further sorted to isolate naïve-Tregs and subsequently transduced with CD19.28z LV and kept in culture in the absence of rapamycin, following the protocol of Imura and coll. (Suppl. Fig. 3a). After 14 days, naïve-derived anti-CD19 CAR-Tregs showed a higher expansion rate compared to Fox19CAR-Tregs (mean fold increase: 436.9 ± 223.1; 19.3 ± 6.3, respectively; p-value < 0.05) (Suppl. Fig. 2b). No differences in transduction efficiency (mean: 64.6% ± 15.3% for naïve-derived CAR-Tregs; 62.2% ± 15.7% for Fox19CAR-Tregs) or Treg purity (mean: 85.5% ± 6.4% for naïve-derived CAR-Tregs; 80.7% ± 6.6% for Fox19CAR-Tregs) were detected at day + 21 (Suppl. Fig. 2c–e).

When cultured with autologous B lymphocytes, Fox19CAR-Tregs proved superior in suppressing the B cell proliferation than naïve-derived CAR-Tregs, especially at lower effector-to-target ratios (p-value < 0.05) (Suppl. Fig. 2f). In terms of cytokine secretion, naïve-derived and Fox19CAR-Tregs showed a similar profile upon antigen-specific stimulation (Suppl. Fig. 2g).

## Fox19CAR LV coupled with rapamycin-based culture efficiently promote a regulatory phenotype in Tconvs

Contaminant Tconvs might affect the purity and ultimately the safety of Treg-based cellular products. On the other hand, by inducing a constitutive FoxP3 expression, the Fox19CAR construct could reprogram Tconvs contaminating the cellular product to a suppressive phenotype. To test this hypothesis, we transduced sorted CD4+CD25- Tconvs with Fox19CAR LV according to two distinct expansion protocols: the Treg protocol, based on IL-2 and rapamycin (IL2R) and the Tconv protocol[35], based on IL-7 and IL-15, in the absence of rapamycin (IL7/15). As negative control, we generated 19CAR-Tconvs starting from CD4+CD25- cells engineered with CAR19.28z LV, and expanded in the presence of IL-7 and IL-15 (Fig. 2a, b).

Transduction efficiency was independent of the expansion protocol used and similar with the two LVs (IL2R Fox19CAR 90.7% ± 2.7%; IL7/15 Fox19CAR 84.2% ± 10.4%, IL7/15 19CAR 83.8% ± 9.0%) (Suppl. Fig. 4a, b). Again, Fox19CAR transduced cells displayed a good co-localization of the two transgenes, independently of the culture conditions (Suppl. Fig. 4c). No differences were observed in the expansion rate promoted by the two protocols or by the vector used (mean fold increase at day 14: IL2R Fox19CAR 20.7 ± 5.6; IL7/15 Fox19CAR 21.9 ± 10.0; IL7/15 19CAR 24.7 ± 7.8, Fig. 2c) but we consistently observed a significantly higher expansion rate in engineered cells compared to the untransduced counterparts (19CAR- p-value < 0.0001 and Fox19CAR-engineered cells p-value < 0.001 both vs UT cells). A large fraction (40%) of transduced cells were characterized by a CD3+CD4+CD25+CD127-FoxP3+ phenotype, independently of the culture conditions and the vector (Suppl. Fig. 4d).

We then verified whether CD4+CD25- cells transduced with the Fox19CAR gained suppressive capacities by culturing autologous PBMCs with IL2R Fox19CAR-, IL7/15 Fox19CAR-, IL7/15 19CAR- or UT

T cells in the presence of a polyclonal stimulus. Only IL2R Fox19CAR lymphocytes efficiently suppressed the proliferation of autologous T cells (mean suppression index 82.3% at 1:1 ratio, Fig. 2d). These results show that the Fox19CAR construct reprograms CD4+CD25- lymphocytes toward a suppressive phenotype and the effect depends on the presence of IL-2 and rapamycin.

## Co-expression of Helios, FoxP3 and TIGIT identifies highly immune suppressive cellular products

We then characterized in better details our Tconv (CD4+CD25-) and Treg (CD4+25+) derived engineered cellular products (Fig. 2a, b) by a multi-parametric flow cytometry panel. CAR-Tconvs and UT counterparts were used as controls. Through a supervised gating strategy, we observed that UT Tregs express higher levels of Glycoprotein A Repetitions Predominant (GARP) than the other cellular products. We found a higher Cytotoxic T Lymphocyte Antigen 4 (CTLA-4) expression on both UT Tregs and Fox19CAR transduced CD4+CD25+ and CD4+CD25- derived cellular products compared to 19CAR-Tconvs, and independently of the culture conditions. Conversely, Glucocorticoid-induced TNFR-related (GITR) protein expression was higher on CAR-Tconvs. Finally, cellular products with proved suppressive capacity displayed a higher expression of T cell Immunoreceptor with Ig and ITIM domains (TIGIT) compared to 7/15 Fox19CAR and 19CAR-Tconvs (Suppl. Fig. 4e).

To capture the complexity of the dataset, we employed cyto-Chain, a software developed in our lab for the unbiased analysis and dimensionality reduction of flow cytometry data[36]. Firstly, we performed a multi-dimensional scaling to identify potential similarities between samples, finding a good separation between regulatory cells and conventional CAR-T or untransduced lymphocytes. Interestingly, IL2R Fox19CAR CD4+CD25- engineered cells localized closer to Tregs than to Tconv cells, in line with the suppressive capacity observed with this Tconv-derived cellular product (Suppl. Fig. 4f). Subsequently, we used the algorithm to classify the different cellular products and thus identified a total of 15 metaclusters. We analyzed more in details the cluster composition of each group and we found 4 differently expressed clusters: cluster 3 and 6 are composed by conventional T lymphocytes while cluster 5 and 12 are constituted by Tregs (Fig. 2e, f).

19CAR- and unmanipulated Tconvs were more represented in cluster 3 and 6. Cluster 3 was composed by CD4+CD25-CD127+FoxP3+ cells, whereas cluster 6 by CD4+CD25-CD127-FoxP3+ ones. Both clusters were characterized by CD4+CD25- elements, indicating the presence of conventional pro-inflammatory T lymphocytes with a high FoxP3 expression due to their activation status. IL7/15 Fox19CAR-Tconvs were more represented in cluster 6, composed by activated Tconvs, thus suggesting their pro-inflammatory nature.

Engineered and unmanipulated Tregs were mainly represented in cluster 5 and 12, comprising CD4+CD25+CD127lowFoxP3+ and CD4+CD25+CD127-FoxP3+ elements, respectively, also characterized by the expression of Helios and TIGIT. These combinations of markers were compatible with a Treg signature. IL2R Fox19CAR-Tconv cells, the only Tconv-derived cellular product with suppressive capacity, were mainly represented in cluster 5, thus supporting their immunosuppressive nature. In addition, both clusters showed an intermediate CD45RA expression on Tregs compared to Tconvs in cluster 3 and 6, compatible with a mixed cell population with a high frequency of effector cells.

Collectively, the unsupervised analysis of flow cytometry data allowed to separate Treg and Tconv derived cellular products, with the exception of the IL2R Fox19CAR CD4+CD25- derived cells, that clusterized with Treg cells, in accordance with the suppressive activity. These findings confirm a Treg profile for suppressive T cells, and indicate a combination of Helios, Foxp3 and TIGIT as a signature associated to the suppressive capacity (Fig. 2e).

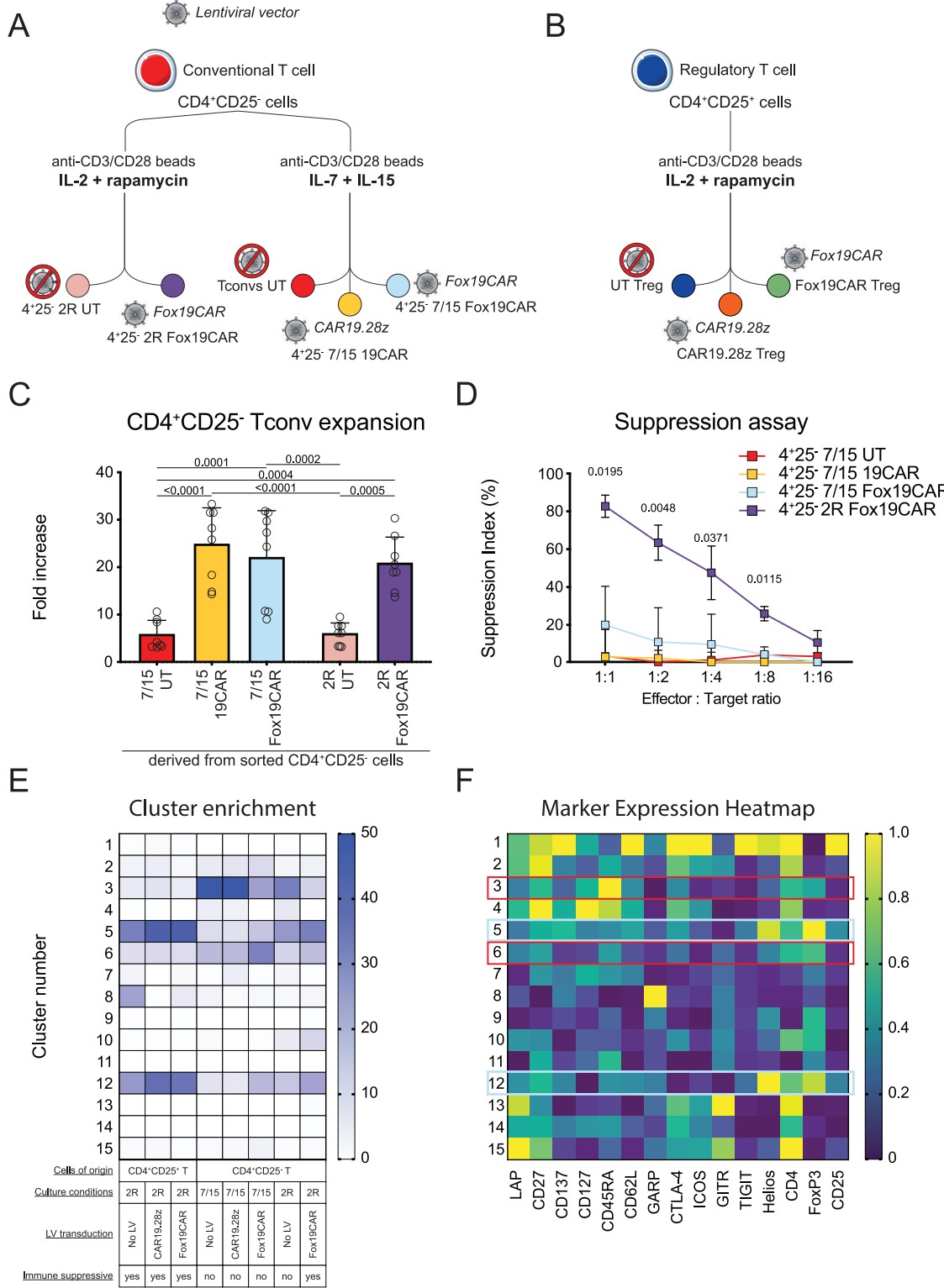

Overall, the combination of a *FoxP3* encoding vector with culture conditions designed for Tregs induces a suppressive phenotype also on Tconvs that might contaminate the initial cellular composition.

### Fox19CAR-Tregs control autoimmunity in vivo
We then turned to test in vivo the functionality and the safety profile of engineered Fox19CAR-Tregs. To do this, we employed a humanized mouse model of SLE that recapitulates chronic inflammatory tissue damage mediated by human immune cells[37]. We transplanted human cord blood stem cells ($0.8-1 \times 10^5$ CD34+ cells/mouse) in irradiated 1 day-old NSG mice. After having established a fully humanized immune system, we i.p. injected mice with pristane, a lipid moiety widely used for experimental SLE induction, since it leads to anti-DNA antibodies formation, lymphopenia and renal damage. Three weeks after, we administered Fox19CAR-Tregs or UT Tregs and analyzed the composition of the bone marrow, spleen, lung and kidney (Fig. 3a).

Circulating human cells were detected 4–5 weeks after humanization and their concentration progressively increased (Fig. 3b, c).

**Fig. 2 | In vitro safety profile of Fox19CAR engineered CD4⁺CD25⁻ T cells.**
**A** Schematic representation of CD4⁺CD25⁻ T cell engineering with different LV vectors and stimulation protocols. **B** Schematic representation of Treg engineering with two different LV vectors. **C** Expansion rate of engineered and untransduced CD4⁺CD25⁻ T cells in the presence of two different cytokine cocktails and transduced with two different LV vectors, assessed at day +14 since the initial stimulation. $N = 8$ for each group. One-way ANOVA with Tukey correction for multiple comparison. The exact $p$-values for each comparison are reported in the graphs. **D** Polyclonal suppressive capacities of either engineered or untransduced CD4⁺CD25⁻ derived T cells. The result is expressed as Suppression index, calculated as Suppression Index = [1-(PBMCs' proliferation with Tregs)/(PBMCs' proliferation alone)] * 100. $N = 3$ for each group. Two-way ANOVA with Tukey correction for multiple comparison. The exact $p$-values for each comparison are reported in the graphs. **E** Heatmap representing the relative abundance of each meta-cluster among different in vitro-expanded CD4⁺CD25⁺ and, CD4⁺CD25⁻ derived cellular products. Fifteen meta-clusters were identified using cytoChain for the unsupervised analysis of flow cytometry data. The darker the color, the higher the abundance of the cluster within the group. $N = 5$. **F** Heatmap reporting the intensity of the evaluated markers in each cluster. Engineered or untransduced Tregs and CD4⁺CD25⁻ T lymphocyte phenotypes were analyzed by flow cytometry. Subsequently, samples were analyzed with cytoChain for an unsupervised scrutiny. For each cluster, the intensity of each marker is reported. Red and cyan boxes highlight the 4 most expressed meta-clusters as reported in Fig. 2E. $N = 5$. All the results are expressed as mean ± standard deviation.

After the appearance of T lymphocytes, at week 7–9, we induced experimental SLE by injection with pristane while maintaining a group of humanized control mice untreated. The treatment was effective, since the animals developed as expected autoantibodies, inflammatory involvement of the lung and the kidney and B cell lymphopenia (Fig. 3b, c and Suppl. Fig. 5a).

Three weeks after pristane administration, mice were randomized to receive either Fox19CAR-Tregs, UT-Tregs ($3.5 \times 10^6$ cells/mouse) or PBS (Suppl. Fig. 5b, c). Fox19CAR-Treg blood concentration peaked 3 days after injection, then decreased at day+7 and remained stable up to day + 10. After 14 days, the pool of engineered cells contracted, and virtually disappeared from the peripheral blood by day + 21 (Fig. 3d). After the injection of PBS or UT-Tregs, we found a progressive and significant reduction of B cells while huCD45⁺ and CD3⁺ T cell counts remained stable (Fig. 3e, f). Mice infused with Fox19CAR-Tregs had stable levels of circulating B cells for up to 10 days (Fig. 3g).

We measured the serum levels of inflammatory and suppressive cytokines before and 3 days after cellular therapy. Fox19CAR-Treg treated mice showed a significant increase in levels of the immune-regulatory agent IL-10, whereas mice treated with UT-Tregs or PBS did not. Levels of inflammatory human cytokines, IFNgamma, IL6, TNFalpha, IL1beta, CCL2, IL17, IL23, IL33 were not significantly increased (Fig. 4a–c and Suppl. Fig. 5d–m).

Collectively, these results indicate that Fox19CAR-Tregs can be safely administered and selectively stabilize B cell counts and delay progression of B cell lymphopenia. Indeed, the stable B cell levels obtained after CAR-Treg injection indicate a limited contaminant CAR-Tconvs if any, being B cell aplasia a hallmark of anti-CD19 CAR-T cell functionality both in patients and in mice[38].

## Restored immune cell composition and reduced inflammation upon Fox19CAR-Treg infusion

To investigate the effects of our cellular products on auto-immunity, we assessed autoantibodies and immune cells composition in the bone marrow and the spleen. Fox19CAR-Tregs significantly reduced the frequency of anti-dsDNA antibody development compared to PBS-injected animals (80% vs 13% of mice treated with CAR-Tregs), while UT Tregs was only partially effective ($p$-value < 0.05) (Fig. 4d). At sacrifice, no differences in terms of relative enrichment in human Tregs or Treg phenotype were found between groups (Fig. 5a and Suppl. Fig. 6a). The percentage of huCD45⁺ leukocytes varied in different organs, but not between treated groups (Fig. 5b, c and Suppl. Fig. 6b). In the spleen of PBS-injected animals, we observed increased frequencies of CD3⁺ and CD4⁺ T lymphocytes compared to non-pristane injected humanized mice ($p$-value < 0.01 for CD3⁺ and CD4⁺ cells). Similar frequencies were observed also in UT Treg animals ($p$-value < 0.001 for CD3⁺ and CD4⁺ cells). Conversely, CAR-Treg mice displayed lower levels of CD3⁺ and CD4⁺ T lymphocytes than the UT Treg ($p$-value < 0.001 for CD3⁺ and CD4⁺ cells) and PBS groups ($p$-value < 0.01 for CD3⁺ and CD4⁺ cells),

comparable to those of non-pristane injected mice (Fig. 5b). A similar situation was detectable for naïve B cells. The CAR-Treg group displayed a significantly higher frequency of naïve B cells than the other two groups of pristane-treated mice, comparable to that observed in non-pristane injected animals (CAR-Treg group vs UT Treg one $p$-value < 0.001). A similar restored homeostasis of the immune cell composition in CAR-Treg treated animals was observed in the bone marrow and kidney (Fig. 5c and Suppl. Fig. 6b). Overall, these data indicate restoration of the immune composition of lymphoid organs upon CAR-Treg infusion.

Experimental SLE in pristane-injected humanized NSG mice was characterized by grossly altered spleen architecture, with multiple granulomatous lesions characterized by giant cells and signs of necrosis. The immune-mediated remodeling abated in Fox19CAR-Treg treated animals, with 60% of them with a normal or negative score ($p < 0.01$). UT-Tregs did not exert significant effects on architecture or inflammatory response in the spleen, where lesions were more numerous than in humanized pristane-treated NSG mice injected with PBS (Fig. 6a, e), suggesting a detrimental role of Tconvs contaminating the cellular product. No significant differences were found in terms of spleen white and red pulp representation, indicating a similar humanization and supporting the flow cytometry data (Fig. 6b).

Similarly, in the lungs of pristane-treated mice an inflammatory reaction characterized by a monocytic/macrophagic infiltrate and granulomatous-like lesions was present. Inflammatory cells localized mainly around blood vessels, also causing their occlusion forming inflammatory thrombi. Alveolar spaces were subverted with the complete loss of the normal architecture. UT-Tregs were ineffective in controlling the inflammatory reaction, even exacerbating it in some mice. Conversely, CAR-Treg mice showed a lower grade of inflammation, with 60% of animals with grade ≤1 lesions and preserved lung structure, normal alveoli and vessel permeability ($p$-value < 0.05) (Fig. 6c, f).

The kidney of pristane-injected animals underwent tubular degeneration and vacuolation, tinctorial changes and cellular sloughing, occasional interstitial and perivascular inflammatory cell infiltrates and sporadic granulomatous inflammation in all mice. Glomerular structures were mostly spared. The kidney remodeling was significantly reduced in Fox19CAR-Tregs treated animals, that displayed significantly less tubular lesions compared to both UT-Treg and PBS-injected animals ($p$-value < 0.01). Actually, the kidney of up to 40% of Fox19CAR-Tregs treated mice had no detectable lesions (Fig. 6d and Suppl. Fig. 6c).

Immunohistochemical analysis of the spleens revealed a prevalent human T cell infiltrate without a clear organization in pristane-treated mice injected with PBS, with only a minimal B cell component. UT Tregs treated animals displayed similar features with a greater B cell infiltrate, in the absence of a clear organization. Conversely, CAR-Treg injection completely restored the normal spleen organization with T and B cells almost exclusively represented inside the white pulp

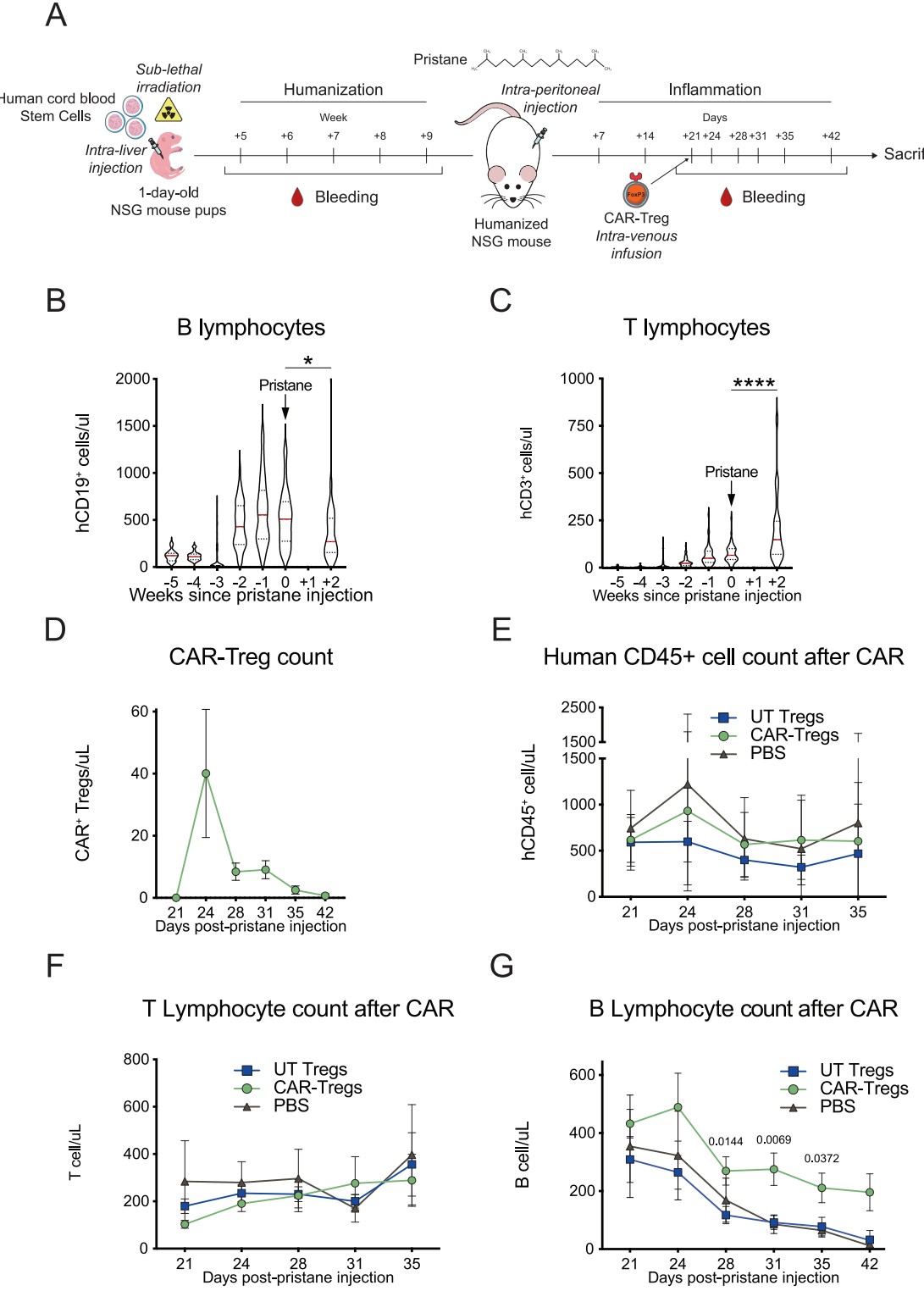

follicles, as in healthy humanized mice (Suppl. Fig. 6d, e). Collectively, these data further support the potent immunoregulatory effect of CAR-Tregs in this SLE model.

**Multiple Fox19CAR-Treg administrations are safe and effective in reshaping the B cell compartment in a prolonged humanized model of autoimmunity**

To explore the possibility of multiple infusions of Fox19CAR-Tregs and to verify their efficacy in a later stage of disease, we adapted the SLE model to SGM-3 mice, a strain characterized by a constitutive high

expression of human IL-3 and GM-CSF, able to better promote human HSC engraftment, and characterized by a longer lifespan than humanized NSG mouse pups[39]. We thus transplanted irradiated 8 weeks-old SGM-3 mice, with human cord blood stem cells ($0.8$-$1 \times 10^5$ CD34 + cells/mouse). After the establishment of a full human immune system, we injected the animals with pristane i.p. to induce the disease. Animals were randomized to receive Fox19CAR-Tregs (3.5 M of cells/mouse) or PBS at 3 and 8 weeks after pristane administration. Additionally, since the efficacy of anti-CD19 conventional CAR-T cells has been recently reported in refractory SLE in patients[23,24], in this model

**Fig. 3 | Efficacy of Fox19CAR-Tregs in an in vivo humanized mouse model of SLE. A** Generation of the humanized mouse model of SLE. 1 day-old pups of NSG mice were irradiated and transplanted with $0.8$-$1 \times 10^5$ human cord-blood stem cells/mouse injected intra-liver. The engraftment was monitored assessing the presence and the composition of human leukocytes on peripheral blood weekly by flow cytometry. After the establishment of a human immune system, mice were injected i.p. with pristane to induce a chronic inflammation. After 3 weeks, UT-, Fox19CAR-Tregs or PBS were injected and their kinetic and the levels of human leukocytes in the peripheral blood were monitored weekly by flow cytometry. Mice were sacrificed 15 weeks after humanization. **B**, **C** Longitudinal assessment of human B and T lymphocytes in peripheral blood. Human B and T cells were identified as huCD45+CD19+ and huCD45+CD3+ lymphocytes, respectively. Absolute cell counts were assessed by flow cytometry. Pristane injection is indicated with an

arrow. $N = 53$ mice. One-way ANOVA with Tukey correction for multiple comparison. *$p$-value 0.021, ****$p$-value < 0.0001. **D** Longitudinal assessment of circulating Fox19CAR-Tregs in mouse peripheral blood after their injection. CAR+ cells were identified with FITC-conjugated recombinant CD19 (rCD19). CAR-Treg absolute counts were assessed by flow cytometry. $N = 16$ mice. **E**–**G** Absolute numbers of circulating human cells in humanized mice in the different groups of treatment after the injection of Fox19CAR-Tregs, UT-Tregs or PBS. Human cells were identified as total huCD45+ leukocytes, huCD45+CD3+ T and huCD45+CD19+ B lymphocytes, respectively. Absolute cell counts were assessed by flow cytometry. $N = 31$ (16 CAR-Tregs, 11 UT-Tregs, 4 PBS). Two-way ANOVA test with Tukey correction for multiple comparisons. The exact $p$-values are reported in the graphs. All the results are expressed as mean ± standard deviation.

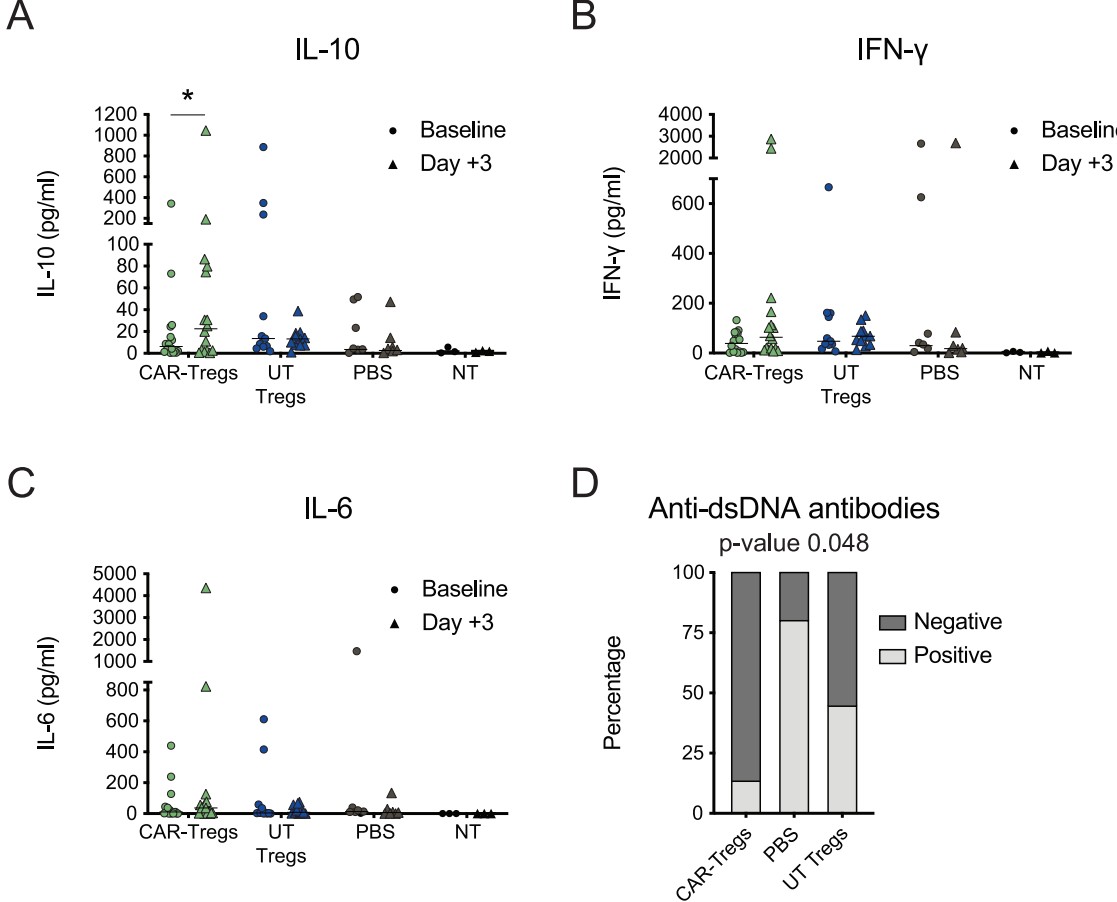

**Fig. 4 | Fox19CAR-Tregs immunomodulatory properties.** Mean IL-10 (**A**), IFN-g (**B**) and IL-6 (**C**) levels in peripheral blood before and 3 days after the injection of Fox19CAR-Tregs, UT-Tregs or PBS. As control, humanized mice not treated with pristane (NT) are included. Cytokine levels were assessed employing a bead-based immunoassay (Biolegend Legendplex 13-plex kit) according to the manufacturer's

instructions. Each dot represents a single mouse. $N = 34$ (16 CAR-Tregs, 11 UT-Tregs, 7 PBS, 3 NT). Two-tailed Wilcoxon test for non-parametric paired data. *$p$-value < 0.025. **D** Frequency of human anti-dsDNA auto-antibodies in mouse serum in the different groups of treatment. $N = 31$ (15 CAR-Tregs, 11 UT-Tregs, 5 PBS). Two-sided Chi-squared test. *$p$-value 0.048.

we also tested the effect of 19CAR-Tconvs (3.5 M of cells/mouse), injected 3 weeks after pristane administration (early therapeutic regimen, Fig. 7a and Suppl. Fig. 7a, b).

Early administered CAR-Tconvs peaked in the peripheral blood at day + 31 and at day +63, then contracted and disappeared by day + 91 (Fig. 7b). As expected, circulating human B cells disappeared in CAR-Tconvs treated mice and remained undetectable until day + 77 (Fig. 7c). Conversely, the administered CAR-Tregs, despite displaying a limited peak of expansion in the peripheral blood at day + 28, induced a significant increase in circulating B cells at day + 35 ($p$-value < 0.05). Starting from day +49, we observed a gradual B cell lymphopenia also in CAR-Treg treated mice, with levels similar to those observed in PBS-

injected animals. Of notice, the second infusion of the CAR-Tregs did not produce detectable CAR-Treg expansion nor did it further increase the B cell count (Fig. 7b, c).

Humanized animals showed a progressive contraction of the circulating huCD45+ and CD3+ cell pools, independently from the group of treatment (Fig. 7d, e). In the CAR-Treg mice, both administrations of engineered cells were associated with a transient increase of total huCD45 + cells. Conversely, CAR-Tconv treated animals showed a delayed transient huCD45 + cell expansion at day + 56, with a peak at day + 63, followed by a progressive reduction, reaching levels comparable to PBS-injected mice by the end of the experiment.

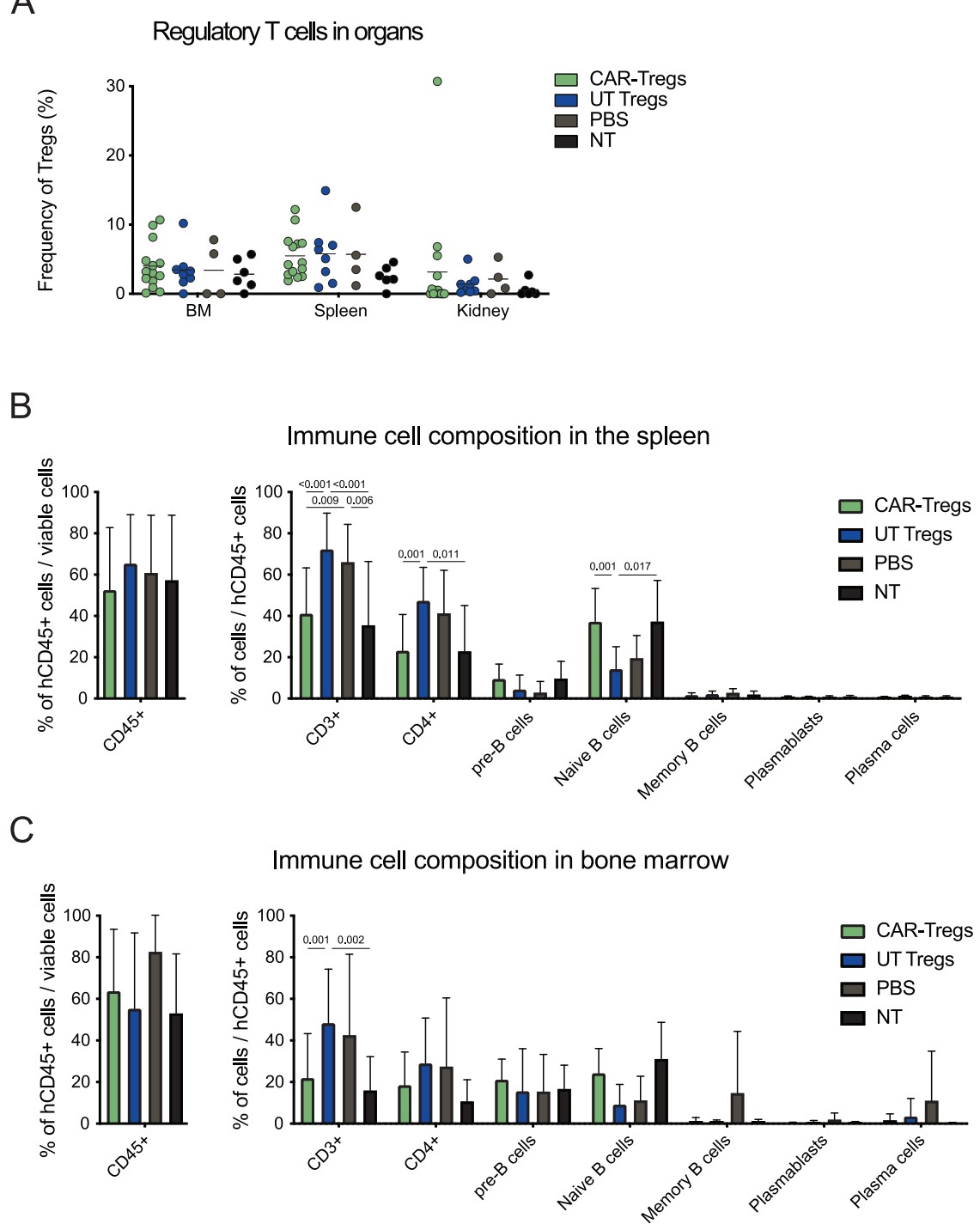

**Fig. 5 | A single injection of Fox19CAR-Tregs restores the human B cell compartment in lymphoid organs. A** Percentage of regulatory T cells in the different organs at sacrifice in the different groups of treatment. Tregs were defined as CD3⁺CD4⁺CD25⁺CD127⁻FoxP3⁺ lymphocytes and their frequency was assessed by flow cytometry. The results are expressed as mean ± standard deviation. As control, humanized mice not treated with pristane (NT) are included. $N = 32$ (14 CAR-Tregs, 8 UT-Tregs, 4 PBS, 6 NT). One-way ANOVA test with Tukey correction for multiple comparisons. **B**, **C** Percentage of total human CD45⁺ cells and human T and B cell sub-populations in the spleen and in the bone marrow in the different groups of treatment at sacrifice. T cells were defined as CD3⁺ cells. B cells were defined as: pre-B cells CD19⁺CD20⁻CD27⁻ cells, naïve B cells CD19⁺CD20⁺CD27⁻ cells, memory B cells CD19⁺CD20⁺CD27⁺ cells, plasmablasts CD19⁺CD20⁻CD27⁺ cells, plasma cells CD138⁺ cells. The results are expressed as mean ± standard deviation. As control, humanized mice not treated with pristane (NT) are included. $N = 38$ (16 CAR-Tregs, 11 UT-Tregs, 5 PBS, 6 NT). One-way ANOVA test with Tukey correction for multiple comparisons. Th exact $p$-values for each comparison are reported in the graphs.

To evaluate the efficacy of engineered cells in a more advanced stage of disease, we compared the effects of Fox19CAR-Tregs or 19CAR-Tconvs injection at 8 weeks after pristane (late therapeutic regimen, Fig. 8a). Engineered CAR-Tconvs and CAR-Tregs showed a rapid expansion, followed by an immediate disappearance from the peripheral blood at day + 66 and +70 after T cell infusion, respectively (Fig. 8b). CAR-Treg treated mice displayed a transient increase in circulating B cells at day + 66 and +70 after T cell infusion, although not statistically significant (Fig. 8c).

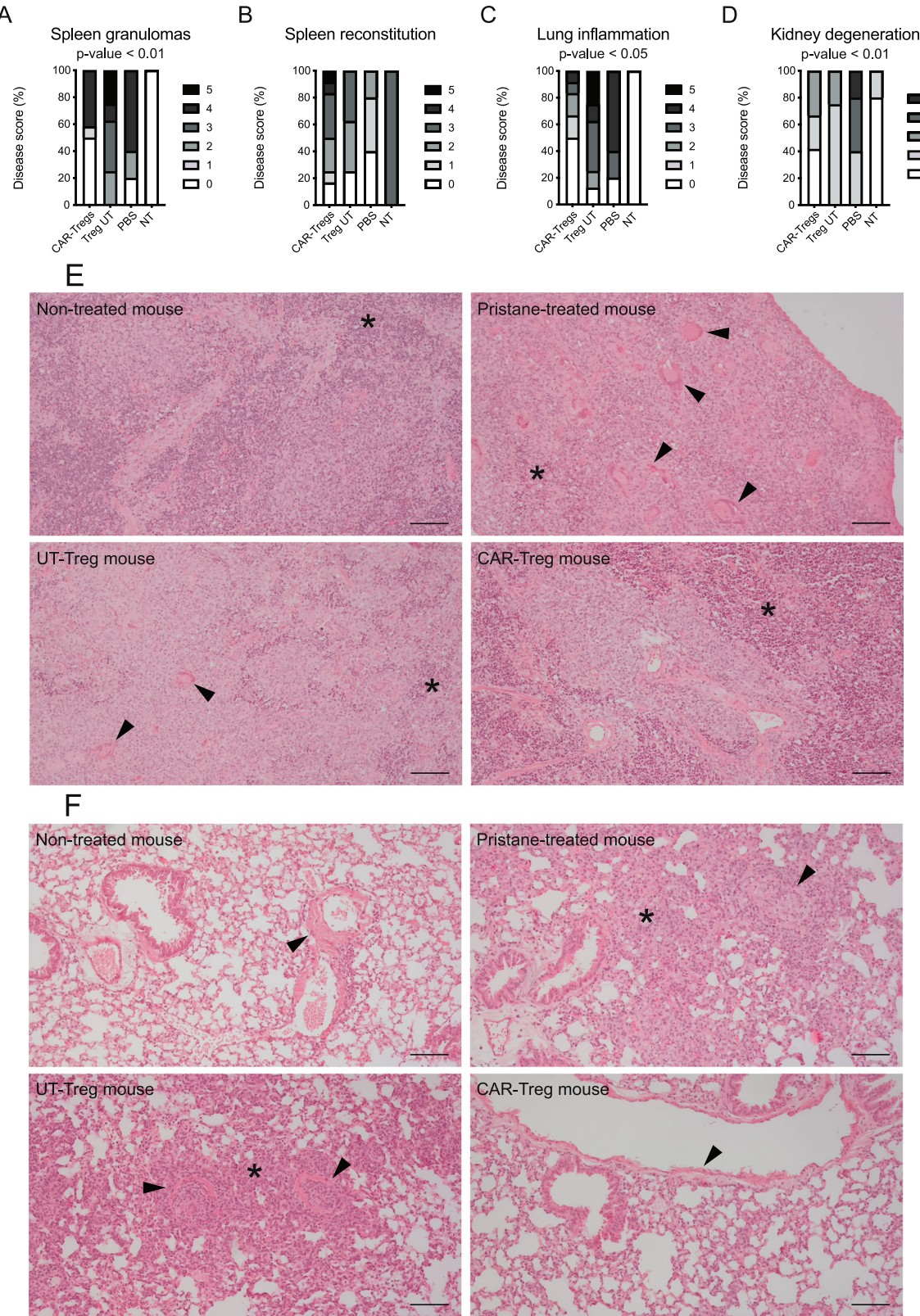

To better elucidate the effects of engineered T cell infusion, we measured the levels of circulating cytokines before and 7 days after each cell administration, in both early and late therapeutic models. In the early therapeutic model, as expected, CAR-Tconv injection was associated with a relevant increase of perforin, Granzyme A and pro-inflammatory cytokines, like IFN-gamma, IL-6 and IL-10 ($p$-value < 0.0001). On the contrary, CAR-Tregs produced high amounts of IL-10

($p$-value < 0.05), IL-2 ($p$-value 0.09) and IL-4, with negligible levels of other pro-inflammatory molecules (Fig. 7f, g). CAR-Treg re-administration 5 weeks after the first injection showed a similar cytokine profile, with a significant increase of IL-10 ($p$-value < 0.03) and a reduction of IL-17 and TNF-alpha (Fig. 8d, e). In the late therapeutic model, we observed a significant increase in IL-10 ($p$-value < 0.0001) with negligible levels of IFN-gamma and IL-6 in CAR-Treg treated mice, whereas

**Fig. 6 | Fox19CAR-Tregs abate inflammatory lesions in different organs.**
**A** Frequency and severity of the inflammatory lesions in the spleen at the sacrifice in the different groups of treatment. As control, humanized mice not treated with pristane (NT) are included. $N = 37$ (16 CAR-Tregs, 11 UT-Tregs, 7 PBS, 3 NT). Two-tailed Chi-squared test. **B** Frequency and grade of human reconstitution in the spleen at the sacrifice, evaluated as representation of red and white pulp. $N = 37$ (16 CAR-Tregs, 11 UT-Tregs, 4 PBS, 6 NT). Two-tailed Chi-squared test. **C** Frequency and severity of the pulmonary inflammatory lesions at sacrifice in the different groups of treatment. As control, humanized mice not treated with pristane (NT) are

included. $N = 37$ (16 CAR-Tregs, 11 UT-Tregs, 4 PBS, 6 NT). Two-tailed Chi-squared test. **D** Frequency and severity of the tubular degeneration in kidneys at the sacrifice. $N = 37$ (16 CAR-Tregs, 11 UT-Tregs, 7 PBS, 3 NT). Two-tailed Chi-squared test. **E** Representative pictures of the spleen in 4 different mice, one per group of treatment. Asterisk indicates the red pulp. Arrowhead indicates areas of granulomatous inflammation. Reference bar 100 μm. **F** Representative pictures of the lungs in 4 different mice, one per group of treatment. Asterisk indicates areas of granulomatous inflammation. Arrowhead indicates blood vessels. Reference bar 100 μm.

CAR-Tconv administration was associated with an increased IFN-gamma production ($p$-value < 0.05) (Fig. 8d, e).

At sacrifice, engineered cells could be detected in the bone marrow with frequencies similar in all groups, while a higher proportion of 19CAR-Tconvs than CAR-Treg was detected in the spleen (Fig. 9a). In bone marrow and spleen, animals injected with an early infusion of 19CAR-Tconv displayed the highest frequency of human CD45+ cells ($p$-value huCD45+ cells early CAR-Tconvs vs. PBS 0.07; early CAR-Tconvs vs. late CAR-Tconvs 0.10; early CAR-Tconvs vs. early CAR-Tregs 0.12), mainly composed by T lymphocytes (%CD3+ cells in the BM early CAR-Tconvs vs. PBS $p$-value < 0.05) (Fig. 9b, c).

In terms of B cells, early CAR-Treg mice showed a significantly higher frequency of CD19 + cells compared to CAR-Tconv and PBS-treated animals both in the bone marrow (early Fox19CAR-Tregs vs. PBS-injected mice $p$-value < 0.05; early Fox19CAR-Tregs vs. late 19CAR-Tconv mice $p$-value < 0.01) and in the spleen (early Fox19CAR-Tregs vs. PBS/early 19CAR-Tconv mice $p$-value < 0.05; early Fox19CAR-Tregs vs. late 19CAR-Tconv / late Fox19CAR-Treg mice $p$-value < 0.01), constituted by pre-/naïve B cells and naïve/memory B cells, respectively. B lymphocytes were still detectable in early 19CAR-Tconv mice, which presented an increased skew towards antibody-producing cells (% plasmablasts in early 19CAR-Tconvs vs. all other groups $p$-value < 0.05 in the bone marrow and the spleen) (Fig. 9d, e).

Pathological evaluation on the bone marrow, the spleen and the liver revealed the presence of an immune cell infiltrate composed mostly by monocytic/macrophagic and histiocytic elements and giant cells, a greatly compromised white pulp reconstitution and a skewing of the hematopoiesis towards myeloid elements with a great abundance of macrophages and eosinophils. Although present in all groups, CAR-Tconv treated animals showed a higher grade of the inflammatory lesions, whereas CAR-Treg and PBS-injected mice appeared similar ($p$-value < 0.05) (Fig. 10a, b). Notably, one out of 5 CAR-Tconv treated mice developed cutaneous GvHD, which on the contrary was absent in all of those that received CAR-Tregs or PBS (Fig. 10c).

Collectively, these results confirm the efficacy of multiple CAR-Treg injections in the early phase of the disease in improving circulating B cells and in reshaping the human immune infiltrate in lymphoid organs. In addition, we confirm the favorable CAR-Treg safety profile, which do not display B cell killing or pro-inflammatory cytokine secretion.

## Discussion

Adoptive cell therapy offers the opportunity to achieve a long-term control of human persistent inflammatory diseases. This might be particularly valuable in SLE, a disease that impacts on patients' survival and quality of life[40] and in which available conventional immunosuppressive agents have substantial limitations[41].

CAR-T technology can be used to enhance the physiological mechanisms that restrain autoimmunity and are defective in SLE patients. To this purpose, conventional T cells expressing a CD19 specific CAR have been successfully tested in preliminary clinical experience[23,24]. Tregs are ideal alternative candidates for such a task. Indeed, in the context of autologous HSCT for autoimmune diseases, long-term disease remission closely relates to the restoration of a

functional Treg compartment, and subsequent immune tolerance[42]. Trials with polyclonal Tregs proved safe but showed limited efficacy in autoimmune diseases[43]. CAR-Tregs demonstrated superior immune suppressive capacities in pre-clinical studies[44]. Considering the role of B cells in the pathogenesis of SLE[45], we decided to seek the proof of principle that non-cytotoxic anti-CD19 CAR-Tregs can be effective by modulating the function of pathogenic self-reactive B lymphocytes.

In our study, we report that anti-CD19 CAR-Tregs acquire indeed new antigen-specific suppressive capacities proving effective against B cells in vitro without detectable inflammatory activity. When employed in vivo in two different humanized mouse model of lupus[37], an early administration of engineered cells did not increase the overall inflammatory burden, reduced the extent of tissue damage, delayed the occurrence of B cell lymphopenia, restored the homeostasis of lymphoid organs while reducing the generation of autoantibodies.

These data confirm that CAR-Tregs exert a potent antigen-specific immunosuppressive activity both in vitro and in vivo. Their specificity for the CD19 B cell antigen did not endow them with the ability to kill B lymphocytes, but rather to exert their regulatory actions at the sites where B cells play their biological role, including the bone marrow, the spleen, and solid organs such as the lung and the kidney. Our results indicate that targeting Treg specificity by CAR to recognize B cells has dramatic effects on the disease natural history, quenching the pathogenic action of the small percentage of disease-causing clones while sparing the vast majority of bystander B cells, that remain able to play their homeostatic role in vivo. Moreover, the pathological evaluation and the flow cytometry analysis on the organs at sacrifice suggest an immunomodulatory activity that might extend beyond B lymphocytes, also involving T cells, probably due to a more extensive loco-regional effect, already described with other CAR-Treg products[27].

As already reported in clinical trials with polyclonal Tregs[46], CAR-Tregs showed a limited persistence. Modifications of the treatment protocol, such as co-administration of CAR-Tregs and low-dose IL-2 or prior lymphodepletion[47,48], might increase Treg survival and persistence[49]. However, in light of the substantial effects observed in our models, we can speculate that an early administration of relatively short-lived CAR-Tregs could anyway play as a molecular switch, interrupting the feed forward cycle by which autoimmunity and tissue damage sustain each other in patients and allowing the restoration of immune homeostasis. This would be an ideal environment for cell therapy, as safety would increase if the desired biological effects could be achieved by transferring cells that do not normally survive for more than a few days.

Tregs don't express specific markers and some of them are shared with conventional T lymphocytes, raising the possibility of contaminant Tconvs in the final cellular product[50]. This aspect is particularly relevant with CAR-Tregs in autoimmunity, where CAR molecules specific for self-antigens are employed and unwanted Tconv transduction could lead to the production of potentially harmful self-reactive CAR-T cells.

Treg sorting strategies have been developed, achieving different grades of purity and cell yields. For a GMP-grade manufacturing, a compromise between feasibility, purity and cell yield is required[33]. We sorted CD4$^+$CD25$^+$ cells as bona fide Treg and used IL2 and rapamycin

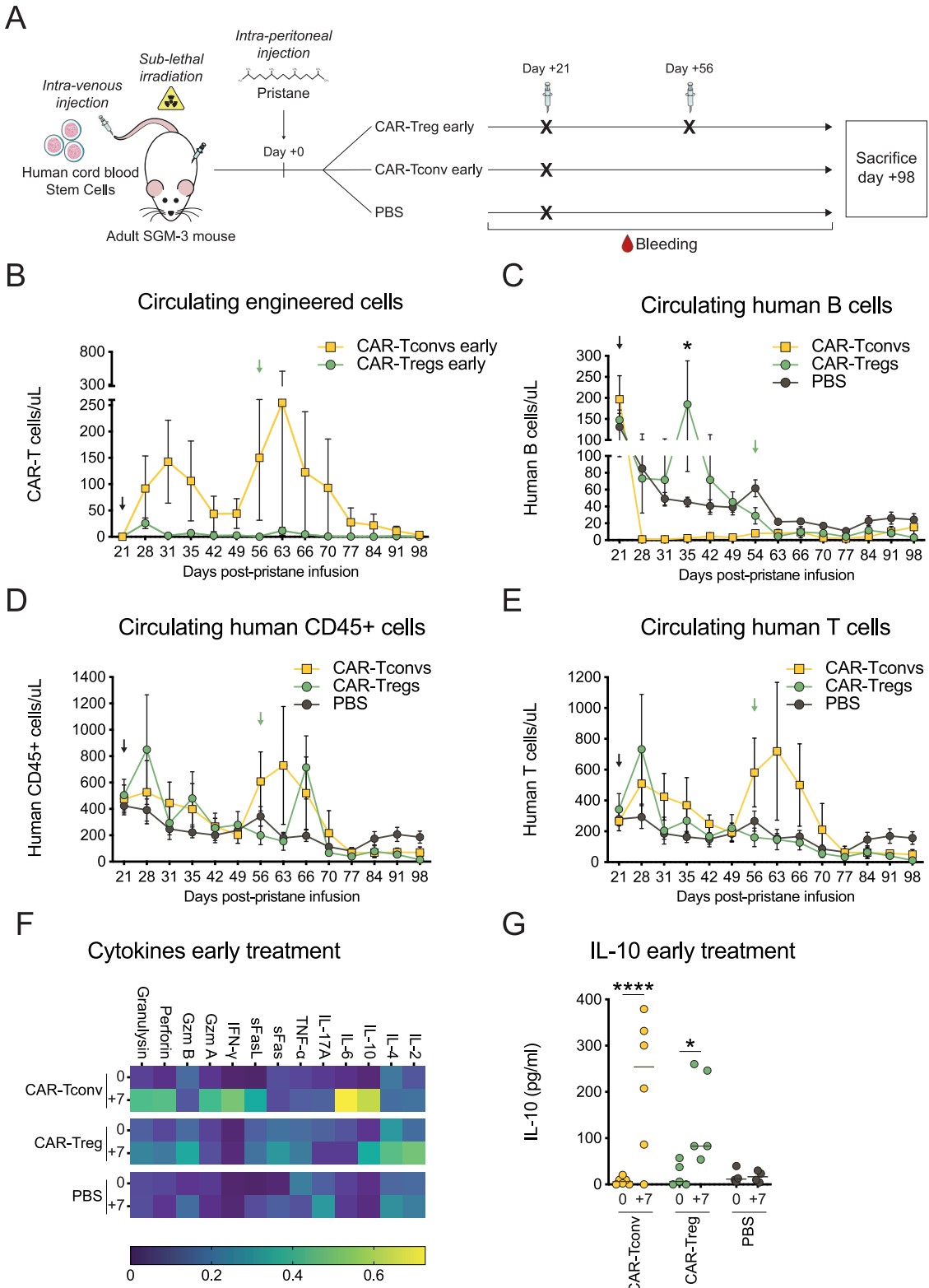

**Fig. 7 | Efficacy and safety of multiple Fox19CAR-Treg injections. A** Schematic representation of the early treatment in a SGM-3 based humanized mouse model of SLE. **B** Longitudinal assessment of circulating Fox19CAR-Tregs and 19CAR-Tconvs after their injection 3 weeks after pristane (early treatment). $N = 11$ (5 CAR-Tregs, 6 CAR-Tconvs). **C–E** Absolute numbers of circulating human cells in the early treatment groups (3 weeks after pristane) after the injection of Fox19CAR-Tregs, 19CAR-Tconvs or PBS. Human cells were identified as total huCD45$^+$ leukocytes, huCD45$^+$CD3$^+$ T and huCD45$^+$CD19$^+$ B lymphocytes, respectively. Absolute cell counts were assessed by flow cytometry. In **B–E**, black arrow indicates engineered cell injection. Green arrow

denotes CAR-Treg second infusion. Results are expressed as mean ± standard deviation. A two-way ANOVA test with Tukey correction for multiple comparisons was employed. $N = 17$ (5 CAR-Tregs, 6 CAR-Tconvs, 6 PBS). *$p$-value 0.029. **F** Cytokine levels in peripheral blood of treated mice, injected 3 weeks after pristane. The amount of each cytokine was normalized across the various groups, scaled to range from 0 (minimum) to 1 (maximum). $N = 17$ (5 CAR-Tregs, 6 CAR-Tconvs, 4 PBS). **G** IL-10 levels in peripheral blood before and 7 days after treatment, 3 weeks after pristane. $N = 14$ (5 CAR-Tregs, 6 CAR-Tconvs, 4 PBS). Two-tailed Wilcoxon test for non-parametric paired data. *$p$-value 0.026, ****$p$-value < 0.0001.

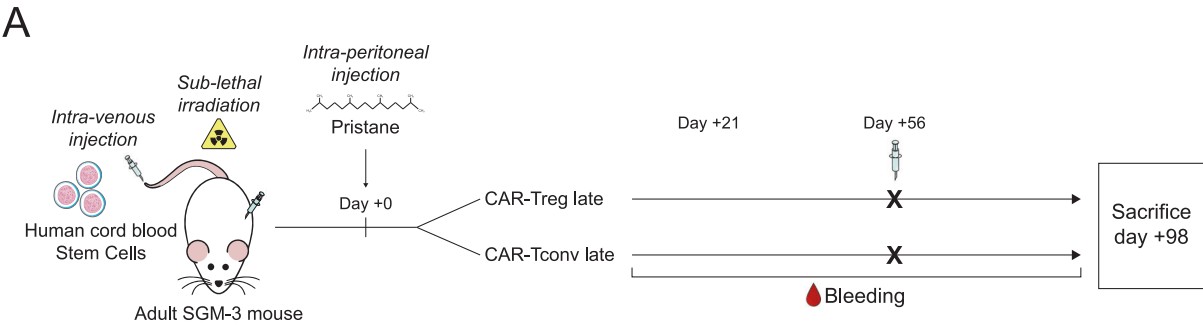

**B**

## Circulating engineered cells

*(Graph: CAR-T cells/uL vs Days post-pristane infusion (56, 63, 66, 70, 77, 84, 91, 98); legend: CAR-Tconvs late, CAR-Tregs late)*

**C**

## Circulating human B cells

*(Graph: Human B cells/uL vs Days post-pristane infusion (56, 63, 66, 70, 77, 84, 91, 98); legend: CAR-Tconvs late, CAR-Tregs late, PBS)*

**D**

## Cytokines late treatment

*(Heatmap: rows IL-2, IL-4, IL-10, IL-6, IL-17A, TNF-α, sFas, sFasL, IFN-γ, Gzm A, Gzm B, Perforin, Granulysin; columns 0/+7 for CAR-Tconv, CAR-Treg, PBS, CAR-Treg reinfusion; color scale 0 to 0.8)*

**E**

## IL10 late treatment

*(Graph: IL-10 (pg/ml) vs CAR-Tconv, CAR-Treg, PBS, CAR-Treg reinf. at 0 and +7; **** and * significance markers)*

in the cell culture to promote Treg expansion without compromising purity, as described[31]. The results confirm that this approach is sustainable, as it meets the requirements for its application in the clinical setting[33].

Tregs may convert to pro-inflammatory Th17 T cells and lose the FoxP3 expression[32]. We hypothesized that a constitutive overexpression of FoxP3 could stabilize the Treg phenotype and restrain contaminant Tconvs. Indeed, FoxP3 overexpression in conventional T lymphocytes converts Tconvs to a suppressive phenotype[34]. We thus included the *FoxP3* gene in the lentiviral vector (Fox19CAR LV). FoxP3 overexpression did not affect Treg expansion or functionality and in combination with IL-2 and rapamycin[51,52] proved effective even in reprogramming CD4$^+$CD25$^-$ T cells (bona fide Tconvs) to a suppressive phenotype. When infused in two different humanized mouse models,

**Fig. 8 | Efficacy and safety of Fox19CAR-Tregs in a late disease stage.**
**A** Schematic representation of the late treatment in a SGM-3-based humanized mouse model of SLE. **B** Longitudinal assessment of circulating Fox19CAR-Tregs and 19CAR-Tconvs after their injection 8 weeks after pristane (late treatment). $N = 10$ (5 CAR-Tregs, 5 CAR-Tconvs). **C** Circulating human B lymphocytes (8 weeks after pristane) after the injection of Fox19CAR-Tregs, 19CAR-Tconvs or PBS. Human B lymphocytes were identified as huCD45+CD19+ cells. Absolute cell counts were assessed by flow cytometry. $N = 16$ (5 CAR-Tregs, 5 CAR-Tconvs, 6 PBS). In **B**, **C**, black arrow indicates engineered cell injection. Results are expressed as mean ± standard deviation. Two-way ANOVA test with Tukey correction for multiple comparisons. **D** Cytokine levels in peripheral blood of treated mice. The amount of each cytokine was normalized, scaled to range from 0 (minimum) to 1 (maximum), and a color gradient was generated. For each group, the relative abundance of each cytokine is reported. $N = 16$ (5 late CAR-Tregs, 3 re-infused CAR-Tregs, 3 late CAR-Tconvs, 5 PBS). **E** IL-10 levels in peripheral blood before and 7 days after treatment 8 weeks after pristane. Results are expressed as mean ± standard deviation. $N = 16$ (5 late CAR-Tregs, 3 re-infused CAR-Tregs, 3 late CAR-Tconvs, 5 PBS). Two-way ANOVA with Tukey correction for multiple comparisons. *$p$-value 0.030, ****$p$-value < 0.0001.

differently from reports with anti-CD19 CAR-Tconvs, Fox19CAR-Tregs did not cause B cell aplasia, Cytokine Release Syndrome (CRS) nor hemophagocytic lymphohystiocytosis (HLH), thus confirming the stability of their immunosuppressive phenotype[53,54]. Interestingly, compared to CAR-Tconvs no signs of xenogenic GvHD were found in CAR-Treg treated mice, in line with the encouraging safety profile of these engineered cells.

The isolation of CD45RA+ naïve Tregs, a purer and more undifferentiated subset, represents an alternative strategy to reduce contaminant Tconvs in Treg-based cellular products. Imura and colleagues proved the efficacy of naïve-derived anti-CD19 CAR-Tregs in suppressing B cell activation both in vitro and in vivo in a xenograft mouse model of GvHD[28]. When directly compared, naïve-derived CAR-Tregs display the highest expansion capacity but Fox19CAR-Tregs showed the highest suppressive activity. Our approach might represent an alternative to Imura's product, displaying an easier manufacturing and a greater flexibility, being employable with either bona fide Tregs or CD4+ Tconvs, potentially improving the scalability in clinical settings where autoimmune patients' lymphopenia might be a limit.

To properly evaluate the efficacy and the safety profile of CAR-Tregs and their interactions with the human immune system, we employed two humanized mouse models of pristane-induced SLE. As reported by Gunawan et al., humanized NSG pups are reliable and faster than immune competent mice in recapitulating some SLE features after pristane injection[37]. However, our humanized mouse models display some limitations. NSG mouse pups have a great inflammatory burden that limits their lifespan and consequently the evaluation of the CAR-Treg long-term efficacy. SGM-3 mice, due to the intrinsic chronic IL-3 and GM-CSF stimulation, have a skewing of the human hematopoiesis toward the myeloid lineage, which ultimately affects the stem cell functionality[39]. In addition, about one-third of SGM-3 mice spontaneously develops HLH after the humanization[55,56]. These two aspects might have affected the engineered cell functionality, potentially limiting their efficacy in SGM-3 mice.

Elegant pre-clinical studies have suggested that conventional anti-CD19 CAR-T cells can be used to selectively deplete B lymphocytes, which play a role in the pathogenesis of SLE, improving the disease in genetic lupus-prone mice[22,25]. More recently, Mackensen et al. showed the efficacy of anti-CD19 CAR-T cells in controlling the disease manifestations in 8 patients with refractory SLE up to 1 year after their injection[23]. Conventional CAR-T cell use, however, might cause CRS or HLH, well-known CAR-T cell toxicities, and theoretically favor opportunistic infections due to the chronic depletion of B cells[38], even though this risk appears highly mitigated in recent studies[57]. Further studies are required to evaluate the long-term efficacy and safety of such a therapy.

Compared to CAR-Tconvs, CAR-Tregs might display a better safety profile. In addition, by exerting an immune suppressive effect directly in lymphoid organs where antigen-presentation and antibody generation occur, anti-CD19 CAR-Tregs might interrupt the vicious cycle that sustains the disease and potentially restore immune tolerance.

Infections, related to an extensive B cell suppression, might represent a potential CAR-Treg side effect. More in general, compared to CAR-Tconvs, the CAR-Treg platform offers the opportunity of safely targeting self-antigens, potentially extending its applicability to other autoimmune diseases where B cell lymphodepletion is ineffective.

In conclusion, here we developed and validated a CAR-Treg cellular product with potent antigen-specific immune suppressive capacities both in vitro and in vivo. In addition, our engineered cells showed an optimal safety profile when employed in two humanized lupus mouse models, proving the stability of the suppressive phenotype. Collectively, these data provide a solid base for the future clinical translation of CAR-Tregs in autoimmunity.

## Methods
This study complies all the required ethical requirements. Healthy donors' Peripheral Blood Mononuclear Cells and CD34+ cord-blood hematopoietic stem cells were employed after the collection of a written informed consent approved by the San Raffaele Ethical Committee. The in vivo studies involving mice were approved by the Italian Ministry of Health.

### Cell lines
Acute lymphocytic leukemia cell line (ALL-CM) was grown in X-VIVO 15 (Lonza) supplemented with 10% FBS (fetal bovine serum, Euroclone), 1% penicillin/streptomycin (Lonza) and 1% glutamine (Lonza). HEK293T and CD40L+ 3T3 cells were grown in IMDM (Lonza) supplemented with 10% FBS (fetal bovine serum, Euroclone), 1% penicillin/streptomycin (Lonza) and 1% glutamine (Lonza). Cells were counted every 2–4 days by Trypan blue dye exclusion according to the cell line and plated at a concentration of $0.5\text{-}1 \times 10^6$ cells/ml.

### Primary T cell isolation and expansion
Peripheral blood mononuclear cells (PBMCs) were collected from healthy donors, after written informed consent, according to the San Raffaele Scientific Institutional Ethical Committee guidelines. PBMCs were isolated by Ficoll-Hypaque gradient separation (Lymphoprep; Fresenius) and were freshly used for the following protocols of specific T cell expansion. After the activation, cells were incubated at 37 °C, 5% CO2, in a humidified cell culture incubator.

### Primary regulatory T cell isolation
CD4+CD25+ regulatory T cells were isolated by magnetic cell separation (Miltenyi), according to manufacturer's instructions. Sorted CD4+CD25+ were activated with cell-sized anti-CD3/anti-CD28 magnetic beads (Dynabeads) in a 3:1 bead: T cell ratio and cultured in X-VIVO 15 (Lonza) supplemented with 10% human serum (Lonza), 1% penicillin/streptomycin (Lonza) and glutamine (Lonza) with Ramapacyn (100 nM). At day 2 after activation, IL-2 (500 UI, Proleukin, Novartis) was added to cell culture and replenished every 2-3 days. After 14 days of cultures, beads were magnetically removed according to the manufacturer's instructions. Starting from day 14, cells were employed for phenotypical evaluation and functional assays.

Naïve Tregs were initially isolated from PBMCs by magnetic cell separation of CD4+CD25+ cells. A subsequent selection of CD127-CD45RA+ elements was done employing a Miltenyi Tyto cell

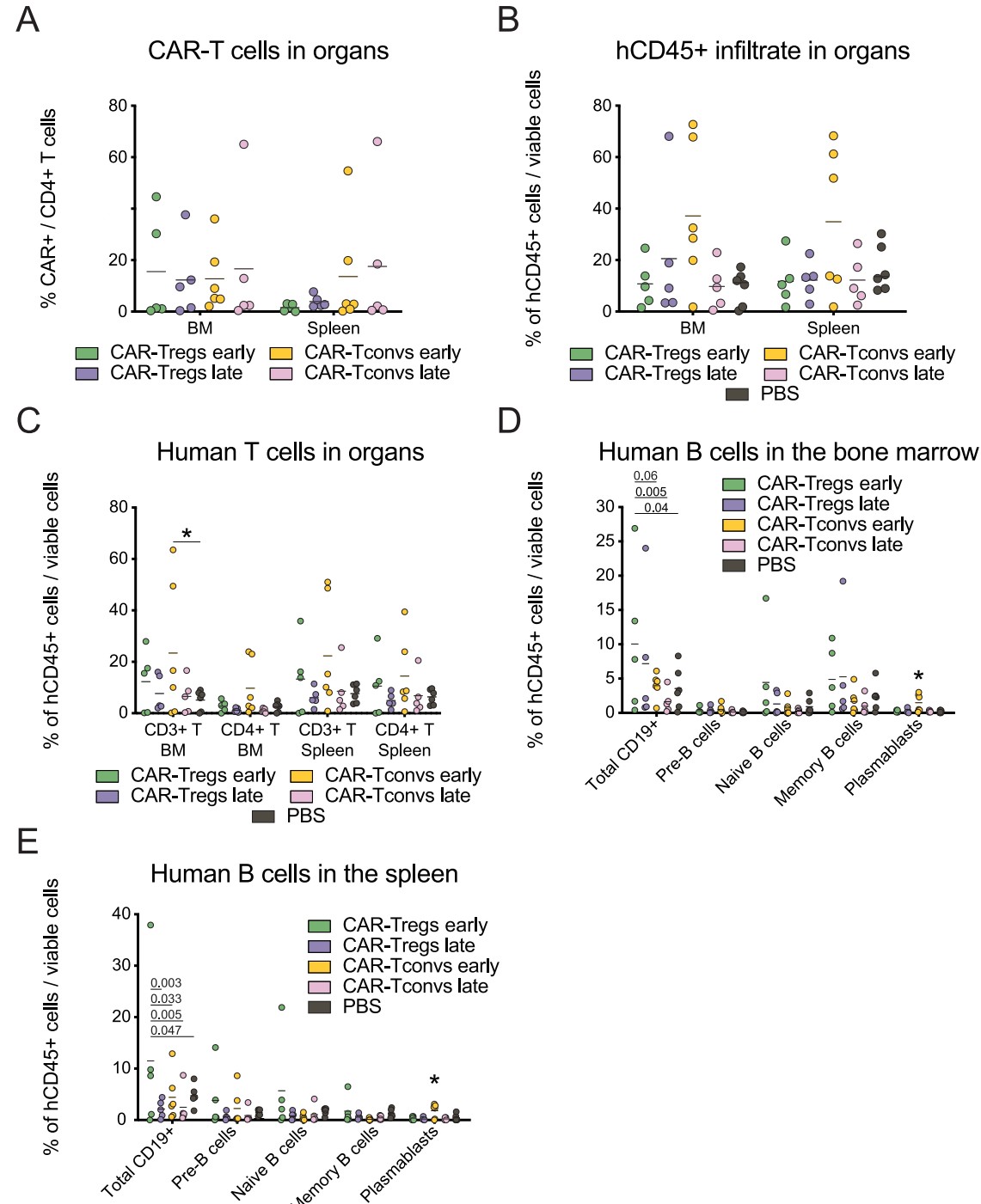

**Fig. 9 | Fox19CAR-Treg efficacy and safety in reshaping the B cell compartment.**
**A** Percentage of CAR⁺ cells in the bone marrow and the spleen at sacrifice. *N* = 21
(5 early CAR-Tregs, 5 late CAR-Treg, 6 early CAR-Tconvs, 5 late CAR-Tconvs).
**B**, **C** Percentage of total human CD45⁺ cells and total T and CD4 + T cells in the bone marrow and the spleen at sacrifice. *p-value 0.043. **D**, **E** Percentage of total B cell sub-populations in the bone marrow and the spleen at sacrifice. In **B**–**E**, results are expressed as mean ± standard deviation. *N* = 27 (5 early CAR-Tregs, 5 late CAR-Tregs, 6 early CAR-Tconvs, 5 late CAR-Tconvs, 6 PBS). One-way ANOVA test with Tukey correction for multiple comparisons. The exact *p*-values for each comparison are reported in the graphs.

sorter. Naïve Tregs were subsequently activated with cell-sized anti-CD3/anti-CD28 magnetic beads (Dynabeads) in a 1:1 bead: T cell ratio and cultured in X- VIVO 15 (Lonza) supplemented with 10% human serum (Lonza), 1% penicillin/streptomycin (Lonza) and glutamine (Lonza) without Ramapacyn. At day 2 after activation, IL-2 (1000 UI, Proleukin, Novartis) was added to cell culture and replenished every 2–3 days. After 9 days, cells were restimulated with anti-CD3/anti-CD28 magnetic beads in a 1:1 bead: T cell ratio. At day + 14 after stimulation,

beads were magnetically removed according to the manufacturer's instructions. Starting from day +14, cells were employed for pheno-typical evaluation and functional assays.

**Primary conventional T cell isolation**
Conventional T Lymphocytes were enriched from isolated PBMCs and stimulated with anti-CD3/CD28 magnetic beads in a 3:1 bead: T cell ratio for 6 days. After 6 days of stimulation, beads were magnetically

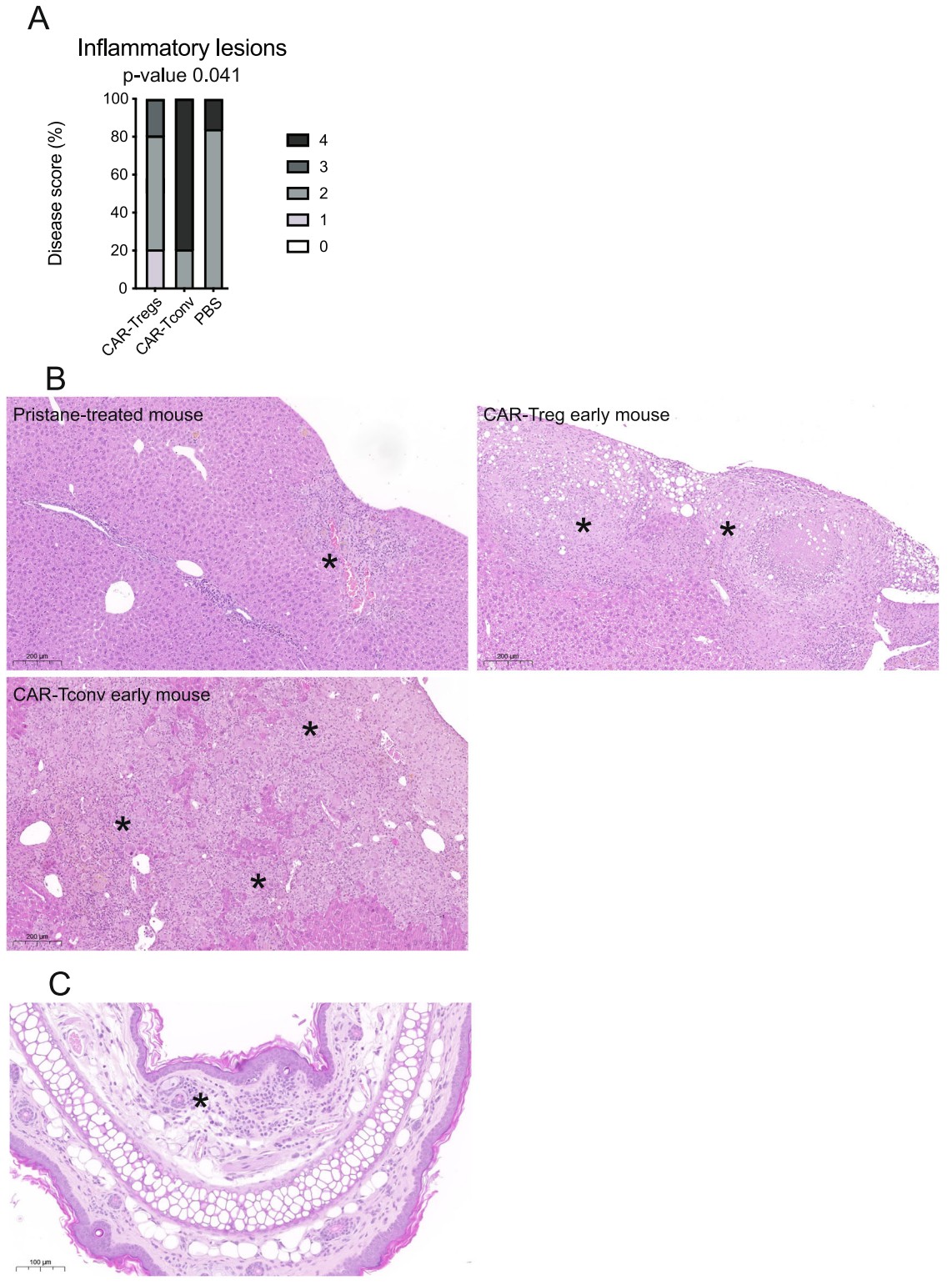

**Fig. 10 | 19CAR-Tconvs worse the inflammatory lesions SGM-3 humanized mice.**
**A** Frequency and severity of the inflammatory lesions at the sacrifice in the different groups of early-administered treatment (3 weeks after pristane). $N = 16$ (5 early CAR-Tregs, 5 early CAR-Tconvs, 6 PBS). Two-tailed Chi-squared test. *$p$-value 0.041. **B** Livers of 3 different mice. Asterisk indicates an area of granulomatous inflammation. Reference bar 200 µm. **C** Ear pinnae of an early-treated CAR-Tconv mouse. Asterisk indicates an area of cutaneous Graft-versus-Host Disease. Reference bar 100 µm.

removed, and cells were maintained in X-VIVO 15 (Lonza) supplemented with 10% human serum (Lonza), 1% penicillin/streptomycin (Lonza) and glutamine (Lonza). T cells were cultured in the presence of IL-7 and IL-15 (5 ng/ml; PeproTech) to preserve early differentiated stem cell memory and central memory phenotype, as previously reported (Cieri et al., 2013). Medium was replaced every 3–4 days and cells were counted by Trypan blue dye exclusion. Starting from day +14, cells were employed for phenotypical evaluation and functional assays.

## CD4$^+$CD25$^-$ T cells

CD4$^+$CD25$^-$ T cells were collected from freshly isolated PBMCs after magnetic cell separation (Miltenyi) according to manufacturer's instructions and were stimulated with cell-sized anti-CD3/anti-CD28 magnetic beads in a 3:1 bead: T cell ratio. Cells were cultured with either IL-7/IL-15 (5 ng/ml, Peprotech) or IL-2 (500 UI, Proleukin, Novartis) and rapamycin (100 nM) in X-VIVO 15 supplemented with 10% human serum (Lonza), 1% penicillin/streptomycin (Lonza) and glutamine (Lonza). Beads were magnetically removed at day +14. Starting from day +14, cells were employed for phenotypical evaluation and functional assays. Medium was replaced every 3–4 days and cells were counted by Trypan blue dye exclusion.

## Viral vectors

**CAR constructs.** CAR19.28z construct was generated by cloning the antigen-specific single chain fragment variable (scFv) into bi-directional lentiviral vector encoding for CAR backbone containing a NGFR-derived mutated-short spacer (NMS) (Casucci et al. 2018), a CD28 transmembrane and co-stimulatory domain and a CD3zeta endodomain under the control of a hPGK promoter. Green Fluorescent Protein (GFP) was cloned in anti-sense under the control of a mCMV promoter. Fox19CAR LV construct was generated by cloning in a unidirectional lentiviral vector both the human FoxP3 gene and an anti-CD19 second-generation CAR under the control of a hPGK promoter. Anti-CD19 CAR was composed by a CD19-specific scFv, a linker peptide derived from the mutated constant portion of IgG4 (Hudechek et al.), the transmembrane and intracellular portion of the CD28 and the CD3z chain. The two transgenes were linked by a thosea-asigna virus 2 A (T2A) peptide. As control, we generated a unidirectional lentiviral vector encoding for the FoxP3 gene under the control of a hPGK promoter.

All employed lentiviral vector backbones were kindly provided by L. Naldini's group.

## LV vector production

Third generation replication-defective and self-inactivating lentiviral vectors (LV) were produced by transient transfection of HEK-293T cells. Briefly, a solution containing the packaging plasmids pMDLg/pRRE (containing HIV-1 gag/pol genes), the pILVV01-Rev plasmid (encoding Rev protein), the envelope plasmid encoding the vesicular stomatitis virus glycoprotein (VSV-G), the pAdvantage (that enhances transient protein expression) and the plasmids carrying the transgene of interest was transfected in sub-confluent 293 T cells using the calcium chloride precipitation method. Supernatants containing lentiviral particles were collected 48 h later and filtered using a 0.22 μm filter, ultracentrifuged at 8000 g for 18 h at 4 °C (Beckman Optima XL-100K Ultracentrifuge), aliquoted and cryopreserved.

## T-cell transduction with LV vectors

At day +2 after stimulation, activated T cells (Treg, Tconv or CD4$^+$CD25$^-$ lymphocytes) from healthy donors were collected and re-suspended at the concentration of $2,5 \times 10^6$ cells/ml in X-VIVO 15. LV vectors were added accordingly to viral titration to achieve an Multiplicity of Infection (MOI) of 10. T cells were kept 24 h at 37 °C and then fresh medium was added in each condition. Transduction efficiency was measured by flow cytometry at day +10/ +14. Recombinant CD19 reagent (Miltenyi) was used for CAR surface staining according to manufacturer's instructions.

## Functional in vitro assays

**Polyclonal suppression assay.** Autologous T lymphocytes (target cells) and engineered T cells (effector cells) were co-cultured at decreasing Effector-to-Target ratio in the presence of anti-CD3/anti-CD28 magnetic beads at 1:10 bead: T cell ratio. To detect proliferating cells, target cells were stained with VioBlue proliferation dye (ThermoFisher), while engineered cells were stained with FarRed

proliferation dye (ThermoFisher), according to manufacturer's instruction. After 6 days, cells were analyzed by flow cytometry. Target and effector cells were discriminated according to the specific proliferation dye. Suppression index was calculated as follows: (1-(percentage of proliferating cells in the effector group)/(percentage of proliferating target cells alone)) x 100.

## Antigen-specific suppression assay

CD19$^+$ B cells (target cells) were negatively sorted from cryopreserved autologous PBMCs using B cell isolation kit II (Miltenyi), according to manufacturer's instruction. To detect proliferating cells, isolated B lymphocytes were labeled with VioBlue proliferation dye (ThermoFisher), according to manufacturer's instruction, whereas engineered T cells (effector cells) were stained with FarRed proliferation dye (ThermoFisher), according to manufacturer's instruction. Labeled target cells and effector cells were co-cultured either in a 1:1 Effector-to-Target ratio in the presence of irradiated CD40L$^+$ 3T3 cells (a fibroblast-derived murine cell line) or at different Effector-to-Target ratios and stimulated with anti-CD40 (0.5 ug/ml), anti-IgM/IgG (20 ug/ml) stimulating antibodies. Cells were analyzed by flow cytometry 3 days later. Suppression Index was calculated as follows: (1-(percentage of proliferating cells in the effector group)/(percentage of proliferating target cells alone)) x 100.

## Antigen-specific killing assay

For antigen-specific killing assay with tumor cell lines, CD19$^+$ ALL-CM cells were co-cultured for 3 days with either engineered or untransduced T cells at 1:5 Effector-to-Target ratio. Cells were analyzed by flow cytometry. The number of residual ALL-CM cells and effector lymphocytes were counted according to the number of elements acquired with flow cytometry and to the total well volume. The anti-tumor activity was expressed as Elimination index as follows: 1−(number of viable target cells in presence of redirected T cells/number of viable target cells in presence of untransduced T cells).

## In vivo humanized models of SLE

**Generation of the models.** The experimental protocol was approved by the Institutional Animal Care and Use Committee (IACUC) with the number 1127. At day 0, 1 day-old NSG mouse (NSG, JAX 005557, RRID IMSR JAX: 005557, Charles-River Italia) pups were sub-lethally irradiated and infused intra-liver with $0.8-1 \times 10^5$ cord-blood derived human CD34$^+$ hematopoietic stem cells. Mice were clinically monitored and weighed every week. Starting from week 5 after cord-blood infusion, the frequency of human cells in the peripheral blood was monitored every week. After the establishment of a full human immune system, chronic inflammation was induced in mice by intra-peritoneal injection of pristane (300 μL, Sigma-Aldrich). A control group of humanized mice did not receive pristane to monitor the dynamics of the immune cells without a chronic inflammatory stimulus. Three weeks after pristane injection, mice were treated with $3.5 \times 10^6$ of anti-CD19 CAR Tregs or UT Tregs or PBS. Human chimerism and human cord-blood derived lymphoid compartment on peripheral blood were assessed weekly by flow cytometry. Mouse weight was monitored every week and, in the presence of weight loss >20%, animals were euthanized. Spleen, bone marrow and kidneys were harvested for flow cytometry and pathology evaluations.

At day 0, 8 weeks-old SGM-3 mice (SGM-3, JAX 013062, RRID IMSR_JAX: 013062, Charles-River Italia) were sub-lethally irradiated and intra-venously infused with $0.8-1 \times 10^5$ cord-blood derived human CD34$^+$ hematopoietic stem cells. Mice were clinically monitored and weighed every week. Starting from week 5 after cord-blood infusion, the frequency of human cells in the peripheral blood was monitored every week. After the establishment of a full human immune system, chronic inflammation was induced in mice by intra-peritoneal injection of pristane (500 μL, Sigma-Aldrich). Mice were

subsequently randomized and divided either in the early or late treatment group. Early treatment group received 3 weeks after pristane injection $3.5 \times 10^6$ of anti-CD19 CAR-Tregs or anti-CD19 CAR-Tconvs or PBS. After 5 weeks (8 weeks after pristane), CAR-Treg treated mice received a second infusion of $3.5 \times 10^6$ of anti-CD19 CAR-Tregs. Late treatment group received $3.5 \times 10^6$ of anti-CD19 CAR-Tregs or anti-CD19 CAR-Tconvs 8 weeks after pristane injection.

SGM-3 human chimerism and human cord-blood derived lymphoid compartment on peripheral blood were assessed weekly by flow cytometry. Mouse weight was monitored every week and, in the presence of weight loss >20%, animals were euthanized by cervical dislocation. Spleen, liver, bone marrow ear pinnae and kidneys were harvested for flow cytometry and pathology evaluations.

For both strains, at euthanasia single-cell suspension from kidney was obtained by using multi-tissue dissociation kit II (Miltenyi) according to manufacturer's instructions with GentleMACS apparatus. Single-cell suspension from spleen was obtained using 70 nm cell strainer. Cells from bone marrow were collected by flushing femurs with PBS. After red-blood cell lysis, cells were suspended in PBS supplemented with 5% FBS. Tregs phenotype and lymphoid immune cells were analyzed by flow cytometry.

All the employed animals were kept in the same animal house under germ-free conditions.

### Anti-dsDNA quantification

The presence of human anti-double-strand DNA (dsDNA) IgG autoantibodies in mouse serum was assessed at euthanasia by immunofluorescence assay according to manufacturer's instruction (QUANTA Lite dsDNA, Werfen Group). A dilution of 1:10 was employed as a threshold for the positivity.

### Pathological evaluation

At euthanasia, mice were euthanized by cervical dislocation. Collected organs fixed in buffered 4% formalin and embedded in paraffin. Haematoxylin and eosin stained 3 µm paraffin sections were examined for histopathological analysis. The evaluation of the specimens was performed in double-blind by a pathologist specifically trained in mouse pathology. We assessed: the normal anatomy, the presence of lesions or fibrosis, the presence and the characteristics of immune cells. For each organ, we evaluated the presence of:

- Spleen: the extension of the red and the white pulp and their relative proportion, the presence and the composition of inflammatory cells and of vasculitis and granulomas
- Liver: the anatomy of the liver and hepatocyte characteristics, the presence of fibrosis, the presence and the composition of the immune infiltrate
- Kidneys: the glomeruli amount and features, the presence of inflammatory cells and of tubular damages/degeneration
- Bone marrow: the presence and the composition of hematopoietic cells and their maturation, the presence of signs of hemophagocytic lympho-histiocytosis
- Ear pinnae: the presence and the composition of skin lesions, the organization and the thickness of the different layers and the presence of Graft-versus-Host Disease.

Microscopic lesions were classified on a scale of 0–5 as minimal (1), mild (2), moderate (3), marked (4), or severe (5); minimal referred to the least extent discernible and severe the greatest extent possible.

Immunohistochemical analysis was performed on selected sections of spleen using rabbit anti-human CD3 (2GV6; Ventana) and mouse anti-human CD20 (L26; Ventana) in automated Ventana Discovery ULTRA system.

### Multiparametric flow cytometry

For Treg phenotype, cells were labeled with titrated fluorescent monoclonal antibodies specific for CD3 (eFluor506, Invitrogen, OKT3, 69-0037-42, 1:75), CD4 (PE-Vio615, Miltenyi, REA623, 130-113-226, 1:150), FoxP3 (PE-Cy5, eBioscience, PCH101, 15-4776-42, 1:1500), CD45RA (BUV496, BD Bioscience, 5H9, 741182, 1:150), CD62L (BUV737, BD Bioscience, SK11, 749210, 1:300), TIGIT (BV786, BD Bioscience, 741182, 747838, 1:100), LAP (Per-CP-eFluor710, eBioscience, FNLAP, 46-9829-42, 1:100), Helios (PE, Biolegend, 22F6, 137216, 1:75), CD25 (PE-Vio770, Miltenyi, REA945, 130-116-205, 1:75), CD27 (APC, Biolegend, M-T271, 356410, 1:150), CD137 (Alexa Fluor 700, Biolegend, 4B4-1, 309816, 1:150), CD127 (APC-Vio770, Miltenyi, REA614, 130-113-416, 1:75), GARP (BV421, BD Bioscience, 7B11, 563956, 1:50), CTLA-4 (BV605, Biolegend, BNI3, 369610, 1:200), ICOS (BV650, BD Bioscience, DX29, 563832, 1:1500), GITR (BV711, Biolegend, 108-17, 371212, 1:400). For 19CAR engineered cells, transduction efficiency was assessed with GFP or NGFR spacer (CD271, BD Bioscience, clone C40-1457), conjugated in BB515 (564580, 1:200), PE (557196, 1:100) or PE-Cy7 (562122, 1:200). Fox19CAR engineered cells were identified using biotinylated recombinant human CD19 reagent (Miltenyi, 130-129-550, 1:67). Biotin was detected using VioBright515-conjugated anti-biotin secondary antibody (Miltenyi, Bio3-18E7, 130-113-298, 1:50). For intra-nuclear staining, lymphocytes were stained with surface antibodies, washed, fixed and permeabilized with FoxP3 staining buffer set (Miltenyi, 130-093-142), according to manufacturer's instructions.

For in vivo experiments, whole blood was lysed with ACK (Ammonium-Chloride-Potassium) buffer for 10 min at room temperature to remove red blood cells. The reaction was then stopped with PBS supplemented with 5% FBS. Subsequently, samples were stained with recombinant CD19 reagent. After washing with PBS supplemented with 5% FBS, fluorochrome-conjugated monoclonal antibodies specific for mouse CD45 (PerCP, Biolegend, 30-F11, 103130, 1:300), human CD45 (PE-Cy7, Invitrogen, HI30, 25-0459-42, 1:300), CD3 (eFluor506, Invitrogen, OKT3, 69-0037-42, 1:150), CD14 (APC-Cy7, Biolegend, 63D3, 367108, 1:300), CD19 (APC, Biolegend, 4G7, 392504, 1:300), CD56 (PE, Biolegend, 5.1H11, 362508, 1:300) were added to samples. Human T cells were counted on peripheral blood using Flow Count fluorescent beads (Beckman Coulter), according to manufacturer's instruction.

For Treg analysis in harvested organs, single-cell suspensions were labeled with fluorochrome-conjugated monoclonal antibodies specific for CD3 (eFluor506, Invitrogen, OKT3, 69-0037-42, 1:75), CD4 (PE-Vio615, Miltenyi, REA623, 130-113-226, 1:150), FoxP3 (PE-Cy5, eBioscience, PCH101, 15-4776-42, 1:1500), CD45RA (BUV496, BD Bioscience, 5H9, 741182, 1:150), CD62L (BUV737, BD Bioscience, SK11, 749210, 1:300), TIGIT (BV786, BD Bioscience, 741182, 747838, 1:100), LAP (Per-CP-eFluor710, eBioscience, FNLAP, 46-9829-42, 1:100), Helios (PE, Biolegend, 22F6, 137216, 1:75), CD25 (PE-Vio770, Miltenyi, REA945, 130-116-205, 1:75), CD27 (APC, Biolegend, M-T271, 356410, 1:150), CD137 (Alexa Fluor 700, Biolegend, 4B4-1, 309816, 1:150), CD127 (APC-Vio770, Miltenyi, REA614, 130-113-416, 1:75), GARP (BV421, BD Bioscience, 7B11, 563956, 1:50), CTLA-4 (BV605, Biolegend, BNI3, 369610, 1:200), ICOS (BV650, BD Bioscience, DX29, 563832, 1:1500), GITR (BV711, Biolegend, 108-17, 371212, 1:400). Recombinant CD19 reagent (Miltenyi, 130-129-550) was used to detect CAR$^+$ T cell, as previously described. Prior to the staining, mouse FC blocking reagent was employed according to the manufacturer's instructions (Miltenyi) to avoid specific binding and to reduce noise.

For immune cells infiltrating the harvested organs at sacrifice, single-cell suspensions were labeled with fluorochrome-conjugated monoclonal antibodies specific for human CD45 (PE-eFluor610, eBioscience, 2D1, 61-9459-42, 1:150) CD3 (eFluor506, Invitrogen, OKT3, 69-0037-42, 1:150), CD4 (BV785, Biolegend, RM4-5, 100552, 1:75), CD19 (FITC, Biolegend, 4G7, 392508, 1:150), CD20 (PE-Cy7, Biolegend, 2H7, 302312, 1:150), CD138 (PE-Cy5, BeckmanCoulter, B-A38, A54191, 1:300), CD27 (APC-Cy7, Biolegend, M-T271, 356424, 1:300), CD14 (PerCP,

Biolegend, M5E2, 301824, 1:150), HLA-DR (BUV805, BD Bioscience, G46-6, 748338, 1:150), PD-1 (BV650, Biolegend, EH12.2H7, 329950, 1:75), CD163 (BV605, BD Bioscience, GHI/61, 745091, 1:300), CD56 (PE, Biolegend, 5.1H11, 362508, 1:300), CD206 (APC, Biolegend, 15-2, 321110, 1:800).

For the in vivo studies, human leukocytes were defined as human CD45+ cells. T cells were defined as huCD45+CD3+ cells. B cell subsets were defined as huCD45$^+$CD19$^+$ lymphocytes. B cell sub-populations were defined as: pre-B cells CD19$^+$CD20$^-$CD27$^-$ cells, naïve B cells CD19$^+$CD20$^+$CD27$^-$ cells, memory B cells CD19$^+$CD20$^+$CD27$^+$ cells, plasmablasts CD19$^+$CD20$^-$CD27$^+$ cells, plasma cells CD138$^+$ cells.

For each experiment, dead cells were excluded by DAPI positive staining. The amount of each antibody has been titrated for each lot as suggested on the data sheet, before the use.

Data were acquired using a BD FACS Canto II or a Symphony A5 (BD Biosciences) and analyzed with FlowJo version 10 software (TreeStar).

### High dimensional flow cytometry analysis
Flow cytometry data were analyzed using cytoChain algorithm for unbiased high-dimensional analysis[36]. Briefly, acquisition stability was evaluated using Flow_iQC algorithm and all the channels' fluorescence intensities transformed by the arcSin function. Successively, the optimized flowSet was analyzed by FlowSOM-based clustering algorithm: the resulting 50 clusters were collapsed into 20 meta-clusters by Consensus Cluster Plus. Marker expressions in each cluster were then organized in a heat map. The frequency of cluster composition in each group was then plotted and analyzed for statistically significant differences using Prism 10 (GraphPad Software).

### Cytokine evaluation
Engineered cells were stimulated in an antigen-specific manner, according to the specific experiment. Culture supernatants were collected after 72 h of stimulation and analyzed with a bead-based assay (Biolegend Legendplex 13-plex kit) according to the manufacturer's instruction by flow cytometry.

Mouse blood samples were collected at the baseline and 3 or 7 days after the cell injection according to the specific experiment. The samples were centrifugated and stored at −20 °C. At the moment of the cytokine analysis, mouse sera were thawed and analyzed with a bead-based assay (Biolegend Legendplex 13-plex kit) according to the manufacturer's instructions by flow cytometry.

### Statistics and reproducibility
Statistical analyses were performed with Prism 10 (GraphPad Software). Student $t$-test was used when comparing two independent groups. Two-way ANOVA was used when comparing three or more independent groups. For non-parametric variables, Mann–Whitney or Kruskall–Wallis tests were employed. $P$-value adjusted tests were employed to identify significant differences between groups. For categorical variables, chi-square test was used. For all comparisons, two-sided $p$-values were used, and $p$-value < 0.05 was considered statistically significant.

Sample size for in vivo studies was calculated with the G-power software to obtain a significant difference in circulating B cells, considering a power of 0.8 and type I error of 0.05. Mice were randomized according to the humanization level and the sex of the animals. In the in vivo experiment involving SGM-3 mice, the randoMice software (Github) was employed to improve this approach, always considering the humanization level and the sex of the animals.

The detailed statistical analysis is reported in the description of each figured.

### Reporting summary
Further information on research design is available in the Nature Portfolio Reporting Summary linked to this article.

## Data availability
The data generated in this study are provided in the Source Data file. Source data are provided with this paper.

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

## Acknowledgements

This work was supported by the Italian Ministry of Research and University (PRIN 2017WC8499), Italian Ministry of Health and Alliance Against Cancer (Ricerca Corrente CAR T project: RCR-2019-23669115), by the EU IMI initiative (T2EVOLVE consortium) to CB. This work was

partially supported by Ministero della Salute (Ricerca Finalizzata Grant No. GR-2016-02364847) to E.R. Parts of the figure were drawn by using pictures from Servier Medical Art. Servier Medical Art by Servier is licensed under a Creative Commons Attribution 3.0 Unported License (https://creativecommons.org/licenses/by/3.0/).

## Author contributions

D.M., U.A., B.B.C. and T.C. contributed with the experimental part, by producing and analyzing the data. C.B. contributed to the execution of the in vivo mouse model. N.R. and S.F. contributed to the pathological evaluation. D.R.S contributed to the analysis of the autoantibodies. C.M. contributed with her expertize in the C.A.R. field to the conception fo the work. G.R., C.F., M.A.A. and B.C. contributed to the critical analysis and review of the data and to the writing of the manuscript.

## Competing interests

BC, CF are inventors on different patents on cancer immunotherapy and genetic engineering. BC has been member of Advisory Board and Consultant for Molmed, Intellia, TxCell, Novartis, GSK, Allogene, Kite/Gilead, Miltenyi, Kiadis, Evir, Janssen and received research support from Molmed s.p.a and Intellia Therapeutics. The other authors declare no competing interests.
