## [Peer Review File · Nature Communications]

Regulatory T cells expressing CD19-targeted chimeric antigen receptor restore homeostasis in Systemic Lupus ErythematosusREVIEWER COMMENTS

Reviewer #1 (Remarks to the Author):

This paper deals with CD19CarTreg therapy in a pristane induced lupus model. The paper describes the generation of the CAR19TRegs using macs-isolated (not sorted) TRegs first and then a different approach in which overexpression of Foxp3 is used to generate CD19 CarTRegs from CD4+CD25 negative T cells, which could simplify the approach to generate huge amounts of TRegs. They then use these Tregs to show their in vitro suppressive capacities and their properties using flow cytometry.

The authors then use a humanized mouse model to show the efficacy of these cells in a pristane induced lupus model.

Major:

The authors demonstrate efficacy of their approach, however what is the conceptual advance compared to the paper of Imura et al (CD19 Car Tregs in GvHD, PMID: 32525846), which is the only CD19CAR TReg paper I could find in Pubmed, but which was unfortunately not even cited and critically discussed by the authors. Imura et al used a humanized GvHD mouse model.

In addition, usually, the pristane model takes 6 months to develop a full blown phenotype. In their work, the authors stop the experiments after 1.5 months. It is unclear, how, with the inherent uncertainties of this model, the data can be explained

Minor:

Fig 1c show flow data and suppression assay at which concentration 1:1?

Fig 1 d and e have been mixed in the text, please show data for killing assays (dots), What does tumor fold increase = 1 mean?

Figure 3: 2 times D

E and F: the meaning of these clusters with respect to the Foxp3CD19CAR Tregs is unclear to me with the explanation delivered in the text. Please explain better, what cluster 3, 4 6 etc. means

Fig 5: Why were only 9 mice tested in figure A (9 mice per group) and 38 in figure B?

What is the cutoff of the anti dsDNA test?

What types of infiltrates are these in the histology? B cells? Any IF or IHC data? If it is T cells, what I would expect in granulomas, how does this relate to CD19CAR Tregs?

Reviewer #2 (Remarks to the Author):

The manuscript of Doglio and colleagues describes the production, the characterization, and the validation of anti-CD19 Chimeric Antigen Receptor (CAR)-expressing regulatory T cells (Treg) for the treatment of systemic lupus erythematosus (SLE). The authors begin with the description of the lentiviral (LV) expression vectors; the first one is expressing CD19CAR and GFP (controlled by separate promoters) and the second is a polycistronic vector encompassing FoxP3 and CD19CAR fused by a 2A-ribosome skipping sequence. The latter is further selected for CD19CAR-Treg validation in different in vitro assays and one in vivo model. In the in vitro assays, the authors show the potency of their CD19CAR Treg to suppress target cells (including activated B cells) and their lack of killing capacities. In addition, the authors tested various production protocols, compared the products, and performed a deep profiling of the CD19CAR Treg. From this, they identified relevant clusters predicting efficient immunosuppressive subsets. The in vivo validation is performed using a remarkable model of humanized mice treated with CD19CAR Treg or non-transduced Treg, confirming that CD19CAR Treg might control SLE while decreasing toxicity. The authors conclude that their data demonstrate the possibility of using CD19CAR as a tool to not kill target cells, but to suppress them. In addition, they oppose their strategy to others using conventional T cells or non-directed Tregs, claiming that specific CD19 targeting also spares normal T cells, helping to maintain a normal B cell population.

There have been previous reports where CD19CAR T cells were evaluated for their capacity to cure SLE (Kansal, 2019, cited in the text). The reviewer did not find any literature concerning the use of CAR-directed Tregs in this context, thus this is novel. However, the authors did not cite the work from Imura and colleagues (2020, PMID: 32525846) which is related to the present manuscript, at least technically. Here the authors use Tregs to block B-cell related GvHD and describe the

manufacture and the validation of CD19CAR Tregs.

Although the experiments are sound, the reviewer feels that some additional tests should be performed, improved in their visualization, or simply repeated to support conclusions. The in vitro evidence is solid concerning the suppressive capacity of the cells; however, the in vivo data are not presented/described in a convincing manner.

Major:

- The statistics are not well described: the type of tests used (Student t-test, ANOVA?) and the exact number of n per group (not total number like in Fig. 5) should be shown in all figures.
- Imura and colleagues (PMID: 32525846) separated Tregs based on their CD45RA/RO levels to avoid losing the immunosuppressive capacity of their product. The authors should discuss and eventually compare their protocol to Imura's. They should also discuss CD45RA expression in the context of their profiling/clustering (Fig. 3).
- CD19CAR expression: the reviewer finds the validation of expression very confusing, and it does not seem that the authors used a direct method to show the presence of CD19CAR, such as anti-Fab or a CD19 chimera, excepted in Fig. 2C. In sup Fig. 1 they monitored the GFP expression, which does not exclude a lack of CAR expression. It would be better to first show that the GFP signal correlates with the NGFR signal, if GFP must be used. It is not clear why the authors did not use rCD19 for all their figures, this could help to assess the CD19CAR expression at the surface.
- Fig. 2 the comparison of the constructs is not fair: first, although the authors use the same promoter for both constructs, one contains a GFP transcribed in opposite direction that could interfere with CD19CAR expression and, second, the CD19CAR molecule designs are not the same (Fig. Sup. 1A and Fig. 2A). Fig. 2B should be either split in two and the staining clearly indicated or replaced with staining with a CD19 chimera, but CAR+ is confusing since different antibodies are used (but only one control is shown). Either the authors show that the level of CD19CAR is the same and that the truncated NGFR does not alter the CAR activity, or they repeat the comparison experiment with a CD19CAR construct exactly the same as the one in the FoxP3 vector. Indeed, this jeopardizes the experiments 2E and 2F where these constructs are compared.
- Fig. 2C, the reviewer would expect that the FoxP3 signal of the LV FoxP3-CD19CAR cells would be a bit shifted compared to the normal one of Tregs (like in Fig. S3C). The expression of FoxP3 should be shown in a more convincing manner.
- Fig. 4D: what happens at D21? Can one really claim a stable level of circulating B cells? The statistics are not clear and maybe a test where the data are paired will tell if the B cell count is stabilized in the CD19CAR Tregs treated animals. Additional experiments should be run to support their conclusions (l. 347-349) because this lymphocyte count cannot rule out an affected B-cell general activity.
- Fig. 4E-G: cytokine levels, which statistical test, n per group? Here again the points should be paired to appreciate the variations. Imura et al. reported some IFN γ in vitro. It could help to repeat the cytokine release in vitro.
- Fig. 5A: Are the groups (N=9, the reviewer concludes that n=3) large enough to say that this is a decrease? This is an important result, and it should be supported by statistics.

Minor:

- The data could be presented in a more harmonious way, especially concerning color code and symbols.
- Figs 1D and 1E seem to be inverted
- What is the effect of FoxP3 alone, the authors should show that FoxP3 only does not affect B cell suppression by T reg (Fig. 2F), this is not essential but would be a fair control
- Fig. 2E: the description is not clear, the authors should show a non-suppressed condition
- Discussion: how do the authors see the introduction of CD19CAR-Tregs in the clinic? Frequency of injection and stability of the product?

Reviewer #3 (Remarks to the Author):

Doglio et al. showed that hCD19-Foxp3-CAR-Tregs suppressed a humanized mouse-SLE model more effectively than non-specific Tregs. The generation of CD19-CAR-Tregs is not novel, however; e.g., Imura et al. JCI insight 2020 (therefore, Figure 1~2 are not novel and should be shown as supplemental figures.)

1. Although hCD19-CAR-Tregs disappeared after administration, suppression of SLE symptoms persisted, suggesting that hCD19-CAR-Treg transfer might be able to induce an immune tolerance. These results are interesting and indicate a unique potential of hCD19-Foxp3-CAR-Tregs for treatment of SLE. However, it is not clear whether the approach is better than hCD19-CAR-T therapy to deplete B cells as SLE treatment. More precise analysis of the mechanism is required.

2. The effectiveness of CD19-CAR T cells in lupus has been reported recently; e.g. Kansal et al. Sci Transl Med, 2019; Jin et al. Cellular & Molecular Immunology 2021; Mackensen et al. Nat Med 2022.

The authors showed that CAR-Treg suppressed lupus in mice and, based on in vitro results (Figure 3), ruled out the possibility the effect could be attributed to a CAR-T activity, although Fig 1D showed that TCD19-CAR-T was superior for B cell inhibition. Further comparison would be required to address the therapeutic effect; for example, whether CD19-CAR-Tregs have a lower killing activity than CD19-CAR-Tconv and thus do not persistently inhibit general B cell responses after treatment.

3. Further analysis is required to address how CD19-Foxp3-CAR-Tregs suppress lupus in humanized mouse model. In Fig. 4E, IL-10+ significantly increased with Treg administration although a small number of mice with high values seem to contribute too much to the result, suggesting that parametric testing may be inappropriate in these experiments. Since IFN-g and IL-6 increased in some mice, it does not seem fair to say that the cytokine profile is suppressed. Figure 4 and 5 can be merged for easier understanding of the experiments.

4. There may be a limit to B cell supply in this humanized model because they are derived from human UCB stem cell transplantation (especially after administration of pristane). This limitation can be discussed. This might contribute to the persistent immune regulation by CAR-Tregs after their disappearance.

Minor comments:

Criteria of histology scoring are not stated. Was scoring performed as double-blind?

The statement about statistical tests is totally insufficient. It should be mentioned individually in each figure legend, not only in Methods section. It is not clear how the multiplicity of tests was handled.

Figure 1D and 1E seemed to be exchanged in figure legend.

Figure 2B; Transduction efficiency should be evaluated in one method as far as they are described in one figure; e.g. CD19 recombinant-Fluor.

REVIEWER COMMENTS

Reviewer #1 (Remarks to the Author):

This paper deals with CD19CarTreg therapy in a pristane induced lupus model. The paper describes the generation of the CAR19TRegs using macs-isolated (not sorted) TRegs first and then a different approach in which overexpression of Foxp3 is used to generate CD19 CarTRegs from CD4+CD25 negative T cells, which could simplify the approach to generate huge amounts of TRegs. They then use these Tregs to show their in vitro suppressive capacities and their properties using flow cytometry. The authors then use a humanized mouse model to show the efficacy of these cells in a pristane induced lupus model.

Reply to Reviewer 1

We thank the Reviewer for summarizing our results. We realized that part of our aims was not completely clear. With this project we aim at developing a strategy to increase both efficacy and safety of CAR-Tregs. For this reason, we designed a bi-cistronic vector to express both the FoxP3 gene and an anti-CD19 CAR construct. We firstly tested this approach on CD4+CD25+ T cells (bona fide Tregs), confirming the CAR-Treg functionality. Subsequently, to ensure that Tconvs potentially contaminating the cellular product would not limit its safety profile, we transduced conventional CD4+CD25- T cells, showing their reprogramming to suppressive cells. Overall, these results indicate that the FoxP3/CAR co-transduction may represent a suitable approach to simultaneously improve the efficacy and the safety of engineered Tregs reported up to date. The goal of the study has been now better clarified in the text.

Reviewer 1, major comment 1

The authors demonstrate efficacy of their approach, however what is the conceptual advance compared to the paper of Imura et al (CD19 Car Tregs in GvHD, PMID: 32525846), which is the only CD19CAR TReg paper I could find in Pubmed, but which was unfortunately not even cited and critically discussed by the authors. Imura et al used a humanized GvHD mouse model.

Reply to Reviewer 1, major comment 1

We thank the Reviewer for raising this issue. Imura and colleagues proved the feasibility of deriving anti-CD19 CAR-Tregs from naïve T cells (CD4+CD25+CD127-CD45RA+ cells) and their efficacy in a model of xeno-GvHD induced by Peripheral Blood Mononuclear Cell (PBMC) injection. We have modified the manuscript by citing this seminal paper both in the Introduction and in the Discussion, critically discussing the pros and cons of each approach. In addition, we systematically compared our cellular product with Imura's one and the results are reported in Suppl. Figure 3. Despite a lower starting population, Imura's cells showed a greater expansion rate compared to our product, probably due to the lack of rapamycin and a greater IL-2 dose in the culture medium. When functionally compared, our cells showed superior antigen-specific suppressive capacities against autologous B cells in 3 independent experiments.

Reviewer 1, major comment 2

In addition, usually, the pristane model takes 6 months to develop a full blown phenotype. In their work, the authors stop the experiments after 1.5 months. It is unclear, how, with the inherent uncertainties of this model, the data can be explained.

Reply to Reviewer 1, major comment 2

We are aware that the classic pristane model, developed in immunocompetent Balb/C or C57BL/6 mice, takes about 6 months to develop. However, when adapted to humanized models in immune compromised mice, the disease develops much more rapidly. In literature, Gunawan et al. (Scientific Reports 2017) reported a humanized mouse model of SLE in NSG mice in which the disease mediated by the human immune system developed 3 months after the administration of pristane.

In our hands, inflammatory features and organ lesions (B cell lymphopenia, granulomas, pneumonia with vasculitis etc.) developed as early as 1 month after the pristane administration. Based on these observations, we stopped for ethical reasons the experiments at 1.5 months.

We ascribe the different kinetics to the sub-lethal irradiation (essential for hematopoietic stem cell engraftment) that might accelerate the development of the disease by increasing the overall inflammatory burden promoted by pristane. The kinetics of the different models have now been mentioned in the Discussion section.

Reviewer 1, minor comment 1

Fig 1c show flow data and suppression assay at which concentration 1:1?

Reply to Reviewer 1, minor comment 1

The Treg : B cell ratio in Figure 1c is 1:1. We modified the figure legend to better specify it. In the new figure disposition, old fig. 1c has been moved to supplementary materials and it has become Suppl. Figure 1h.

Reviewer 1, minor comment 2

Fig 1 d and e have been mixed in the text, please show data for killing assays (dots), What does tumor fold increase = 1 mean?

Reply to Reviewer 1, minor comment 2

We thank the Reviewer for this comment. We modified the text according to the figure order and we added the dots representing single data to the figure. Tumor fold increase is the ratio between the number of tumor cells alone or in the presence of either CAR-Tregs, UT Tregs and conventional CAR-T cells divided by the number of tumor cells alone, as detailed in the legend to Supplemental Figure 1 of the revised manuscript. It reflects tumor cell increase or reduction in the investigated condition.

Reviewer 1, minor comment 3

Figure 3: 2 times D

Reply to Reviewer 1, minor comment 3

We thank the Reviewer for this annotation. We corrected the figure accordingly.

Reviewer 1, minor comment 4

E and F: the meaning of these clusters with respect to the Foxp3CD19CAR Tregs is unclear to me with the explanation delivered in the text. Please explain better, what cluster 3, 4, 6 etc. means.

Reply to Reviewer 1, minor comment 4

We thank the Reviewer for this comment. We have rephrased the paragraph, explaining in better detail the composition and the meaning of each cluster.

Reviewer 1, minor comment 5

Fig 5: Why were only 9 mice tested in figure A (9 mice per group) and 38 in figure B?

Reply to Reviewer 1, minor comment 5

We thank the Reviewer for raising this question. We have increased the number of analyzed sera and we have updated the figure and the figure legend accordingly.

Reviewer 1, minor comment 6

What is the cutoff of the anti dsDNA test?

Reply to Reviewer 1, minor comment 6

Anti-dsDNA antibodies were measured by IFA methods and the cutoff was a dilution of 1:10. The information has been added to Material and Methods Section.

Reviewer 1, minor comment 7

What types of infiltrates are these in the histology? B cells? Any IF or IHC data? If it is T cells, what I would expect in granulomas, how does this relate to CD19CAR Tregs?

Reply to Reviewer 1, minor comment 6

We thank the Reviewer for raising this point. To better clarify the characteristics of the immune infiltrate, we analyzed by immunohistochemistry spleens of mice from the various groups. Results revealed both B and T cells in granulomas. In mice treated with CAR-Tregs fewer granulomas were formed compared to all other groups of mice, with restored organization of lymphoid organ structure, possibly due to a more extensive loco-regional effect by CAR-Tregs, as already described with other CAR-Treg products (DOI: 10.1016/j.jaut.2018.05.003). We reported the results in the text and in Suppl. figure 6G and H. These findings are in line with those already described by employing flow cytometry and hematoxylin-eosin staining.

Reviewer #2 (Remarks to the Author):

The manuscript of Doglio and colleagues describes the production, the characterization, and the validation of anti-CD19 Chimeric Antigen Receptor (CAR)-expressing regulatory T cells (Treg) for the treatment of systemic lupus erythematosus (SLE). The authors begin with the description of the lentiviral (LV) expression vectors; the first one is expressing CD19CAR and GFP (controlled by separate promoters) and the second is a polycistronic vector encompassing FoxP3 and CD19CAR fused by a 2A-ribosome skipping sequence. The latter is further selected for CD19CAR-Treg validation in different in vitro assays and one in vivo model. In the in vitro assays, the authors show the potency of their CD19CAR Treg to suppress target cells (including activated B cells) and their lack of killing capacities. In addition, the authors tested various production protocols, compared the products, and performed a deep profiling of the CD19CAR Treg. From this, they identified relevant clusters predicting efficient immunosuppressive subsets. The in vivo validation is performed using a remarkable model of humanized mice treated with CD19CAR Treg or non-transduced Treg, confirming that CD19CAR Treg might control SLE while decreasing toxicity. The authors conclude that their data demonstrate the possibility of using CD19CAR as a tool to not kill target cells, but to suppress them. In addition, they oppose their strategy to others using conventional T cells or non-directed Tregs, claiming that specific CD19 targeting also spares normal T (B) cells, helping to maintain a normal B cell population.

There have been previous reports where CD19CAR T cells were evaluated for their capacity to cure SLE (Kansal, 2019, cited in the text). The reviewer did not find any literature concerning the use of CAR-directed Tregs in this context, thus this is novel. However, the authors did not cite the work from Imura and colleagues (2020, PMID: 32525846) which is related to the present manuscript, at least technically. Here the authors use Tregs to block B-cell related GvHD and describe the manufacture and the validation of CD19CAR Tregs.

Although the experiments are sound, the reviewer feels that some additional tests should be performed, improved in their visualization, or simply repeated to support conclusions. The in vitro evidence is solid concerning the suppressive capacity of the cells; however, the in vivo data are not presented/described in a convincing manner.

Reply to Reviewer 2

We are delighted to learn that the Reviewer finds our work novel and based on an informative model. We have now answered all issues raised by the Reviewer and we believe that the manuscript is significantly improved.

Reviewer 2, major comment 1

The statistics are not well described: the type of tests used (Student t-test, ANOVA?) and the exact number of n per group (not total number like in Fig. 5) should be shown in all figures.

Reply to Reviewer 2, major comment 1

We thank the Reviewer for this comment. We modified the figure legends and the Materials and Methods section to better describe the statistical analysis.

Reviewer 2, major comment 2

Imura and colleagues (PMID: 32525846) separated Tregs based on their CD45RA/RO levels to avoid losing the immunosuppressive capacity of their product. The authors should discuss and eventually compare their protocol to Imura's. They should also discuss CD45RA expression in the context of their profiling/clustering (Fig. 3).

Reply to Reviewer 2, major comment 2

We thank the Reviewer for having raised this issue. We systematically compared our cellular product with Imura's one and the results are now reported in Suppl. Figure 3. Imura's cells showed a greater expansion rate compared to our product, probably due to the lack of rapamycin and a greater concentration of IL-2 in the culture medium. However, when functionally compared, our cells showed superior antigen-specific suppressive capacities against autologous B cells in three independent experiments. We have modified the manuscript by critically discussing the pros and cons in the Discussion section. In addition, we included a discussion about the CD45RA expression in our cellular product, also considering the results obtained with the use of an unbiased algorithm for the flow cytometry data analysis.

Reviewer 2, major comment 3

CD19CAR expression: the reviewer finds the validation of expression very confusing, and it does not seem that the authors used a direct method to show the presence of CD19CAR, such as anti-Fab or a CD19 chimera, excepted in Fig. 2C. In sup Fig. 1 they monitored the GFP expression, which does not exclude a lack of CAR

expression. It would be better to first show that the GFP signal correlates with the NGFR signal, if GFP must be used. It is not clear why the authors did not use rCD19 for all their figures, this could help to assess the CD19CAR expression at the surface.

Reply to Reviewer 2, major comment 3

We thank the Reviewer for this comment. The CD19CAR construct is a bi-directional lentiviral vector encoding for the CAR transgene in sense and GFP in antisense. The CAR transgene contains a truncated LNGFR moiety as spacer (Casucci et al, *Frontiers in Immunology* 2018). We validated the CAR detection with two different methods: anti-NGFR antibody and Miltenyi recombinant CD19 (rCD19) obtaining comparable results. Once verified CAR expression and functionality, we tested the correlation between GFP and CAR expression, that proved well balanced, as already observed with other transgenes in bi-directional lentiviral vectors (Amendola et al *Nat Biotech* 2005). These data have been formatted in the revised version of the manuscript in Suppl. Figure 1E. After this validation, we employed only rCD19 to detect the CAR.

Reviewer 2, major comment 4

Fig. 2 the comparison of the constructs is not fair: first, although the authors use the same promoter for both constructs, one contains a GFP transcribed in opposite direction that could interfere with CD19CAR expression and, second, the CD19CAR molecule designs are not the same (Fig. Sup. 1A and Fig. 2A). Fig. 2B should be either split in two and the staining clearly indicated or replaced with staining with a CD19 chimera, but CAR+ is confusing since different antibodies are used (but only one control is shown). Either the authors show that the level of CD19CAR is the same and that the truncated NGFR does not alter the CAR activity, or they repeat the comparison experiment with a CD19CAR construct exactly the same as the one in the FoxP3 vector. Indeed, this jeopardizes the experiments 2E and 2F where these constructs are compared.

Reply to Reviewer 2, major comment 4

As mentioned above, we used a bi-directional LV, in which the 2 transgenes are controlled by the same promoter. The vector has been reported able to promote robust and well-balanced transgene expression (Amendola et al *Nat Biotech* 2005) and this characteristic has been confirmed in T cells by our team (Provasi, Genovese et al., *Nat. Med.* 2012). In the revised Figure (now Supplementary Figure 2B), CAR expression promoted by the 2 vectors was compared using the same staining method, i.e. rCD19.

Reviewer 2, major comment 5

Fig.2C, the reviewer would expect that the FoxP3 signal of the LV FoxP3-CD19CAR cells would be a bit shifted compared to the normal one of Tregs (like in Fig. S3C). The expression of FoxP3 should be shown in a more convincing manner.

Reply to Reviewer 2, major comment 5

We thank the Reviewer for this comment. We did not observe an increase in Foxp3 expression in Tregs transduced with the FoxP3-CD19CAR construct (*Figure 1 for Reviewer*). This result could be due to natural high levels of FoxP3 expression typical of Regulatory T cells. In these cells, intrinsic mechanisms finely tune the expression and the amount of the FoxP3 gene/complex. This reflects in a maximum Foxp3 expression (and consequently maximum MFI) that is not further boosted upon transduction in those cells with an intrinsic high expression. Similar results were also reported by Allan et al. *Molecular Therapy* 2008 (DOI: [10.1038/sj.mt.6300341](https://doi.org/10.1038/sj.mt.6300341)). We think that the real advantage of the FoxP3 transduction is obtained on the FoxP3-cells that could potentially contaminate our products or in those cells with an unstable expression.

Figure 1 for Reviewer's perusal

The figure reports the FoxP3 Mean Fluorescence Intensity (MFI) in either Treg UT or Fox19CAR-Tregs at day +21 assessed by flow cytometry. Black lines represent the mean. N = 6 per group.

Reviewer 2, major comment 6

Fig. 4D: what happens at D21? Can one really claim a stable level of circulating B cells? The statistics are not clear and maybe a test where the data are paired will tell if the B cell count is stabilized in the CD19CAR Tregs treated animals. Additional experiments should be run to support their conclusions (l. 347-349) because this lymphocyte count cannot rule out an affected B-cell general activity.

Reply to Reviewer 2, major comment 6

We included in Figure 4 the levels of circulating B cells at day +21. A paired test confirmed the stability of circulating B cells after CAR-Treg administration. We included in the figure legend a detailed description of the statistical analysis.

Reviewer 2, major comment 7

Fig. 4E-G: cytokine levels, which statistical test, n per group?

Reply to Reviewer 2, major comment 7

We employed a paired Wilcoxon test to compare the levels of each cytokine between the baseline and 3 days after cell injection in each group. We modified the figures to explicit statistical method and numbers.

Reviewer 2, major comment 8

Here again the points should be paired to appreciate the variations. Imura et al. reported some IFN γ in vitro. It could help to repeat the cytokine release in vitro.

Reply to Reviewer 2, major comment 8

We thank the Reviewer for raising this issue. In the revised version of the manuscript we compared Fox19CAR- and naïve-derived anti-CD19 CAR Treg cytokine secretion upon antigen-specific stimulation and we did not find any significant difference among these two populations. Compared to untransduced Tregs, engineered cells produce less IFN-gamma, suggesting the presence of a Tconv population in the first group of cells.

We also confirmed the reduced production of pro-inflammatory cytokines by Fox19CAR-Tregs in a different humanized mouse model of SLE (Figure 6 and Suppl. Figure 7). When injected at different time points in humanized SGM-3 mice treated with pristane, we found that Fox19CAR-Tregs produced high amount of IL-10 with negligible levels of IFN-gamma and IL-6, thus confirming the immunoregulatory properties of these cells.

Reviewer 2, major comment 9

Fig. 5A: Are the groups (N=9, the reviewer concludes that n=3) large enough to say that this is a decrease? This is an important result, and it should be supported by statistics.

Reply to Reviewer 2, major comment 9

We have increased the number of analyzed sera and we have updated the figure and the figure legend accordingly.

Reviewer 2, minor comment 1

The data could be presented in a more harmonious way, especially concerning color code and symbols.

Reply to Reviewer 2, minor comment 1

We have substantially modified the figures, and employed a uniform color code (each color representing a vector) and symbol code (each symbol representing a cell of origin) throughout the manuscript.

Reviewer 2, minor comment 2

Figs 1D and 1E seem to be inverted

Reply to Reviewer 2, minor comment 2

We thank the Reviewer for this annotation. We corrected the figure accordingly.

Reviewer 2, minor comment 3

What is the effect of FoxP3 alone, the authors should show that FoxP3 only does not affect B cell suppression by T reg (Fig. 2F), this is not essential but would be a fair control.

Reply to Reviewer 2, minor comment 3

To address this question, we generated a new LV construct with the FoxP3 gene alone and we transduced Tregs with it. We added these data to Figure 1 and we modified the text to include this section. When directly compared with Fox19CAR-Tregs, FoxP3-transduced Tregs did not show significant antigen-specific capacities.

Reviewer 2, minor comment 4

Fig. 2E: the description is not clear, the authors should show a non-suppressed condition

Reply to Reviewer 2, minor comment 4

We thank the Reviewer for this observation. The results are expressed in terms of suppression index, a ratio between the Tconv proliferation in the presence and in the absence of Tregs, ranging from 0 (no suppression) to 1 (total suppression) that is calculated as follows:

Suppression index = $1 - \frac{\text{Tconv proliferation with Tregs}}{\text{Tconv proliferation alone}}$

We have modified the figure legend to better clarify better the assay. We also have included a non-suppressed condition to the figure, as suggested.

Reviewer 2, minor comment 5

Discussion: how do the authors see the introduction of CD19CAR-Tregs in the clinic? Frequency of injection and stability of the product?

Reply to Reviewer 2, minor comment 4

We thank the Reviewer for raising this important issue. We modified the discussion to extend our perspective for this important topic.

Reviewer #3 (Remarks to the Author):

Doglio et al. showed that hCD19-Foxp3-CAR-Tregs suppressed a humanized mouse-SLE model more effectively than non-specific Tregs. The generation of CD19-CAR-Tregs is not novel, however; e.g., Imura et al. JCI insight 2020 (therefore, Figure 1~2 are not novel and should be shown as supplemental figures.)

Reply to Reviewer 3

We have now modified the manuscript to better underline the innovative aspects of the project and its difference with Imura's approach. We moved Figure 1 in the supplementary material (now Suppl. Figure 1), as suggested by the Reviewer. In figure 2 we demonstrate the functionality of a novel lentiviral vector, coupling a CAR construct and the FoxP3 gene, not reported by Imura et al. that is novel and, in our opinion, is worth being maintained in the principal figures (now Figure 1). Indeed, when directly compared to Imura's cells, our cell product showed superior antigen-specific activity and potentially increased clinical feasibility (Suppl. Figure 2, revised discussion).

Reviewer 3, major comment 1

Although hCD19-CAR-Tregs disappeared after administration, suppression of SLE symptoms persisted, suggesting that hCD19-CAR-Treg transfer might be able to induce an immune tolerance. These results are

interesting and indicate a unique potential of hCD19-Foxp3-CAR-Tregs for treatment of SLE. However, it is not clear whether the approach is better than hCD19-CAR-T therapy to deplete B cells as SLE treatment. More precise analysis of the mechanism is required.

Reply to Reviewer 3, major comment 1

We thank the Reviewer for raising this issue. To properly compare CAR-Tregs with CAR-Tconvs in different stages of the disease, we performed a second experiment in SGM-3 adult mice, a strain engineered to express high levels of IL-3 and GM-CSF, which have an increased lifespan, thus potentially allowing to better assess the long-term efficacy of our treatment. We reported the results of this experiment in Figure 5 and Suppl. Figure 6. Early administered CAR-Tconvs peaked in the peripheral blood at day +10 and at day +42, then contracted and disappeared by day +70. As expected, circulating human B cells disappeared in CAR-Tconvs treated mice and remained undetectable until day +56. Conversely, the administered CAR-Tregs, despite displaying a limited peak of expansion in the peripheral blood at day +7, induced a significant increase in circulating B cells at day +14 (p -value <0.05). Starting from day +28, we observed a gradual B cell lymphopenia also in CAR-Treg treated mice, with levels similar to those observed in PBS-injected, (pristane treated) animals. In CAR-Tconvs treated animal we observed worsening of organ lesions, not observed in CAR-Treg treated animals, and the development of xenogeneic GvHD in 1 out of 6 treated animals. In this model, we verified the effects of a second infusion and a delayed infusion of CAR-Tregs. None of these infusions produced relevant clinical results, possibly due to the intrinsic chronic IL-3 and GM-CSF stimulation present in SGM-3 mice, and consequent skewing of the human hematopoiesis toward the myeloid lineage, which ultimately affects the stem cell functionality (DOI: 10.1038/sj.leu.2403222). We included in the revised manuscript a discussion about the limits of the SGM-3 mouse model.

Reviewer 3, major comment 2

The effectiveness of CD19-CAR T cells in lupus has been reported recently; e.g. Kansal et al. Sci Transl Med, 2019; Jin et al. Cellular & Molecular Immunology 2021; Mackensen et al. Nat Med 2022. The authors showed that CAR-Treg suppressed lupus in mice and, based on in vitro results (Figure 3), ruled out the possibility the effect could be attributed to a CAR-T activity, although Fig 1D showed that TCD19-CAR-T was superior for B cell inhibition. Further comparison would be required to address the therapeutic effect; for example, whether CD19-CAR-Tregs have a lower killing activity than CD19-CAR-Tconv and thus do not persistently inhibit general B cell responses after treatment.

Reply to Reviewer 3, major comment 2

We thank the Reviewer for this comment. When tested *in vitro*, 19CAR-Tconvs only killed autologous B cells while 19CAR-Tregs restrained the B cell proliferation without killing the target (new Suppl figure 1). As mentioned, to compare CAR-Tregs and CAR-Tconvs, we performed an experiment in adult humanized SGM-3 mice, confirming the good CAR-Treg safety profile, already obtained with humanized NSG pups (Figure 4/5 and Suppl. Figure 6). Engineered Treg cells secreted immunoregulatory cytokines without IFN-gamma and IL-6 and reshaped the B cell compartment without showing antigen-specific killing capacities. We added these findings also in the Discussion.

Reviewer 3, major comment 3

Further analysis is required to address how CD19-Foxp3-CAR-Tregs suppress lupus in humanized mouse model. In Fig. 4E, IL-10+ significantly increased with Treg administration although a small number of mice with high values seem to contribute too much to the result, suggesting that parametric testing may be inappropriate in these experiments.

Reply to Reviewer 3, major comment 3

We thank the Reviewer for this comment. Statistical analysis in fig. 4E with a paired Wilcoxon test to compare the levels of each cytokine between the baseline and 3 days after cell injection in each group confirmed the significance of the difference.

Reviewer 3, major comment 4

Since IFN-g and IL-6 increased in some mice, it does not seem fair to say that the cytokine profile is suppressed.

Reply to Reviewer 3, major comment 4

We modified the text accordingly.

Reviewer 3, major comment 5

Figure 4 and 5 can be merged for easier understanding of the experiments.

Reply to Reviewer 3, major comment 5

We would like to express our gratitude to Reviewer 3 for the valuable input regarding the organization of the figures to enhance reader comprehension of the experimental strategy. While we acknowledge the suggestion to merge Figures 4 and 5 for ease of interpretation, we firmly believe that the results of the pathological evaluation are not only quite striking but also critically important for a thorough scientific assessment of our work. As a result, we have decided to maintain these figures (Figure 3 and Figure 4) as separate entities within the revised manuscript.

Reviewer 3, major comment 6

There may be a limit to B cell supply in this humanized model because they are derived from human UCB stem cell transplantation (especially after administration of pristane). This limitation can be discussed. This might contribute to the persistent immune regulation by CAR-Tregs after their disappearance.

Reply to Reviewer 3, major comment 6

We thank the Reviewer for raising this question. Although with some differences compared to humans, it is reported in literature that humanized mouse models in NSG mice can efficiently recapitulate B cell differentiation and IgG class switch (doi: [10.1093/infdis/jit448](https://doi.org/10.1093/infdis/jit448)). We think that humanized mice are reliable systems in which studying human B cells and their dynamics. In support of this, the feasibility of such a model in recapitulating B cell immune responses against infectious diseases in specific strains like SGM-3 or NRG has been demonstrated (doi: [10.3389/fimmu.2018.02734](https://doi.org/10.3389/fimmu.2018.02734)). Inflammatory conditions, similar to those we produce upon pristane injection, can further boost B cell differentiation and the production of antigen-specific immunoglobulins (doi: [10.1182/blood-2016-04-709584](https://doi.org/10.1182/blood-2016-04-709584)).

Reviewer 3, minor comment 1

Criteria of histology scoring are not stated. Was scoring performed as double-blind?

Reply to Reviewer 3, minor comment 1

We thank the Reviewer for this comment. The evaluation of the specimens was performed in double-blind by a pathologist specifically trained in mouse pathology. We modified the method section, to specify the criteria adopted for the pathological evaluation. Briefly, we assessed: the normal anatomy, the presence of lesions or fibrosis, the presence and the characteristics of immune cells. For each organ, a score ranging from 0 to 5, considering all these features was created and employed:

- Spleen: the extension of the red and the white pulp and their relative proportion, the presence and the composition of inflammatory cells and, eventually, of vasculitis and granulomas
- Kidneys: the glomeruli amount and features, the presence of inflammatory cells and of tubular damages/degeneration
- Bone marrow: the presence and the composition of hematopoietic cells and their maturation, the presence of signs of hemophagocytic lympho-histiocytosis

Reviewer 3, minor comment 2

The statement about statistical tests is totally insufficient. It should be mentioned individually in each figure legend, not only in Methods section. It is not clear how the multiplicity of tests was handled.

Reply to Reviewer 3, minor comment 2

We modified all figure legends to better describe the employed statistical analysis.

Reviewer 3, minor comment 3

Figure 1D and 1E seemed to be exchanged in figure legend.

Reply to Reviewer 3, minor comment 3

We thank the reviewer for this annotation. We corrected the figure legend according to the figure disposition.

Reviewer 3, minor comment 4

Figure 2B; Transduction efficiency should be evaluated in one method as far as they are described in one figure; e.g. CD19 recombinant-Fluor.

Reply to Reviewer 3, minor comment 4

We thank the Reviewer for this comment. To solve our shortcoming, we added a flow cytometry plot to demonstrate the correspondence between NGFR, rCD19 and GFP, reported in Suppl. Figure 1E. Starting from figure 2 (now figure 1), we employed only rCD19 to detect the CAR.

REVIEWER COMMENTS

Reviewer #1 (Remarks to the Author):

The authors addressed all major issues and added another humanized mouse model with repeated injections of CAR T Regs to the paper.

Reviewer #2 (Remarks to the Author):

The authors performed a substantial amount of work including an additional animal experiment containing a comparative group treated with clinically used CAR T cells. There are still some points which deserve better clarification.

Major comment 3 and 4:

Although the reviewer understands the explanations of the authors and rather appreciates the efforts to clarify this topic. It still feels that the level of rCD19 binding should be clearly presented for all CD19CAR expressing T cells in parallel with a given experiment. It is not normal practice that the reader has to look for explanation/justification of correlation when a "universal" label is available. Technically all experiments comparing CD19CAR in different backgrounds or setups should show rCD19 staining to support that the CAR levels are comparable and do not affect the interpretation of the results. It is also not acceptable to validate level of expression based on previous reports, all CAR scientists know that the CAR presence at the membrane is extremely variable from donor-to-donor and between experiments.

The reviewer recommends to reduce the staining demonstration to only rCD19 data, but to show it for each comparative experiment.

Major comment 9:

Fig. 4A: the number of animals was increased and the antibody quantification confirms/improves the first result. Although not a statistician, the reviewer is wondering if the use of Chi-square(d) test is valid here. In addition, the explanations in the text are not clear, the authors should describe what is observed: is there a difference between UT Treg and the other groups?

Minor comment 1:

The presentation is improved, just check supFig1B CAR Treg are in the same colour as Tconv., on the graph the UT Treg are blue but final product is red/orange?.

Minor comments from 2nd round:

- F1EF: Harmonize the scales (y-axis)
- F3BC: Use the same numbering as in A on the scale (x-axis), it is confusing to do the math while reading the figure
- Lines 103-104: Rephrase, Imura also talked about B-cell pathology
- Line 138: Numbers not clear and do not correspond to SF1I
- Line 157: 2A is a ribosome skipping sequence, not "self-cleaving" (although often use, even in Wikipedia) this is not a cleavage but a separation
- Lines 178-179: this can be said only if the authors show equal CD19CAR levels in all groups with the cells used in this experiment
- Lines 203-204: F2D, is it really a safety test, if the problem is B cell aplasia, shouldn't the authors show a killing experiment? Please comment on this.

Reviewer #3 (Remarks to the Author):

The authors have properly revised the manuscript according to the reviewers' comments.

REVIEWER COMMENTS

Reviewer #1 (Remarks to the Author):

The authors addressed all major issues and added another humanized mouse model with repeated injections of CAR T Regs to the paper.

Reply to Reviewer #1:

We are delighted that Reviewer #1 appreciated and found exhaustive the work done in the revision phase.

Reviewer #2 (Remarks to the Author):

The authors performed a substantial amount of work including an additional animal experiment containing a comparative group treated with clinically used CAR T cells. There are still some points which deserve better clarification.

Reply to Reviewer #2:

We thank the Reviewer for the comment.

Reviewer #2, major comment 3 and 4:

Although the reviewer understands the explanations of the authors and rather appreciates the efforts to clarify this topic. It still feels that the level of rCD19 binding should be clearly presented for all CD19CAR expressing T cells in parallel with a given experiment. It is not normal practice that the reader has to look for explanation/justification of correlation when a "universal" label is available. Technically all experiments comparing CD19CAR in different backgrounds or setups should show rCD19 staining to support that the CAR levels are comparable and do not affect the interpretation of the results. It is also not acceptable to validate level of expression based on previous reports, all CAR scientists know that the CAR presence at the membrane is extremely variable from donor-to-donor and between experiments.

The Reviewer recommends to reduce the staining demonstration to only rCD19 data, but to show it for each comparative experiment.

Reply to Reviewer #2, major comment 3 and 4:

As suggested, we produced multiple graphs showing the CAR expression, in terms of both percentage and Relative Fluorescence Intensity (RFI), assessed with rCD19 for each functional experiment involving both CAR19- and Fox19CAR-Treg/T cells, demonstrating the comparable expression level among the various cellular products. We included the data in the various supplementary figures, next to each functional experiment.

Reviewer #2, major comment 9:

Fig. 4A: the number of animals was increased and the antibody quantification confirms/improves the first result. Although not a statistician, the reviewer is wondering if the use of Chi-square(d) test is valid here. In addition, the explanations in the text are not clear, the authors should describe what is observed: is there a difference between UT Treg and the other groups?

Reply to Reviewer #2, major comment 9:

We thank the Reviewer for the comment. The data about the autoantibody positivity are categorical variables with only two levels of response (positive or negative). Chi-square test represent the best option for such a kind of variables. Unfortunately, this statistic test is unable to perform multiple comparisons, highlighting only the "overall" difference among groups and allowing to affirm that the observed difference is significant (rejection of the zero hypothesis). Conversely, multiple comparisons among the different subgroups would introduce potential p-value overestimation errors.

We performed a Fisher exact test for 2x2 tables to test the statistical difference in any combination possible with the three groups (CAR-Tregs, UT-Tregs and PBS). CAR-Tregs showed a significant difference when compared with PBS) p-value <0.01) but not with the UT-Treg group. Conversely, there was no difference among UT-Tregs and PBS.

We have rephrased the sentence in the text to better clarify this point.

Reviewer #2, minor comment 1:

The presentation is improved, just check supFig1B CAR Treg are in the same colour as Tconv., on the graph the UT Treg are blue but final product is red/orange?

Reply to Reviewer #2, minor comment 1:

We thank the Reviewer for the comment. In Supp. Fig. 1B blue denotes untransduced Tregs, red indicates untransduced conventional T cells and orange refers to 19CAR-Tregs.

Reviewer #2, minor comment 2:

Minor comments from 2nd round: F1EF: Harmonize the scales (y-axis)

Reply to Reviewer #2, minor comment 2:

We have adjusted the scale of the graphs as suggested.

Reviewer #2, minor comment 3:

F3BC: Use the same numbering as in A on the scale (x-axis), it is confusing to do the math while reading the figure

Reply to Reviewer #2, minor comment 3:

We have adjusted the x-axis of the graphs in Fig. 3 and Supp. Fig. 5 as suggested, referring everything to the pristane injection considered as week 0.

Reviewer #2, minor comment 4:

Lines 103-104: Rephrase, Imura also talked about B-cell pathology

Reply to Reviewer #2, minor comment 4:

We have rephrased the sentence as suggested.

Reviewer #2, minor comment 5:

Line 138: Numbers not clear and do not correspond to SF11

Reply to Reviewer #2, minor comment 5:

We have modified the sentence to include the percentage of proliferating B cells in each group, as shown in Supp. Fig. 11.

Reviewer #2, minor comment 6:

Line 157: 2A is a ribosome skipping sequence, not "self-cleaving" (although often use, even in Wikipedia) this is not a cleavage but a separation

Reply to Reviewer #2, minor comment 6:

We have removed the self-cleaving word from the sentence.

Reviewer #2, minor comment 7:

Lines 178-179: this can be said only if the authors show equal CD19CAR levels in all groups with the cells used in this experiment

Reply to Reviewer #2, minor comment 7:

As answered in major comment 3 and 4, the various CAR products showed comparable levels of CAR expression detected with rCD19. For this reason, we kept the sentence as written.

Reviewer #2, minor comment 8:

Lines 203-204: F2D, is it really a safety test, if the problem is B cell aplasia, shouldn't the authors show a killing experiment? Please comment on this.

Reply to Reviewer #2, minor comment 8:

We thank the Reviewer for this comment. In our experience and as shown in the results regarding the in vitro experiments with CAR-Tregs, we have found a strong correlation between the suppression assay results obtained with either a polyclonal or an antigen-specific stimulation and the killing capacities of the tested cellular product. For this reason, even if not directly tested, we think that the evaluation of the overall suppressive capacities of engineered CD4⁺CD25⁻ - derived cells is predictive of their regulatory rather than killing capacities in the presence of the antigen.

Reviewer #3 (Remarks to the Author):

The authors have properly revised the manuscript according to the reviewers' comments.

Reply to Reviewer #3

We are delighted the Reviewer #3 appreciated and found exhaustive the work done in the revision phase.

REVIEWERS' COMMENTS

Reviewer #2 (Remarks to the Author):

The reviewer is satisfied with the authors' response and has no more comments. He thanks them for their careful corrections.

REVIEWER COMMENTS

Reviewer #2 (Remarks to the Author):

The reviewer is satisfied with the authors' response and has no more comments. He thanks them for their careful corrections.

Reply to Reviewer #2:

We are delighted that Reviewer #2 appreciated and found exhaustive the work done in the revision phase.